# Objective-Behavior Alignment: Diagnostics for MORL Policy Selection

## Abstract

Real-world decision-making often requires optimizing multiple competing objectives simultaneously. In reinforcement learning (RL), this is typically addressed by combining reward signals into a single scalar objective via a scalarization function, which can be fragile: small changes in the weights can induce drastically different policies. Multi-objective reinforcement learning (MORL) instead produces sets of policies that explicitly represent trade-offs between objectives. However, these policies are typically presented to the decision maker only through their value vectors, which can obscure substantial behavioral variation: policies that induce distinct trajectories may appear indistinguishable when evaluated solely by expected returns. We propose an exploratory diagnostic workflow that automatically highlights behavioral variation along the Pareto front that objective values alone do not reveal, providing both quantitative and visual tools to support policy inspection. We validate our approach on simple grid examples and scale it to continuous control benchmarks, demonstrating that it remains effective as problem complexity increases.

## 1 Introduction

Real-world scenarios often require optimizing for several objectives at once (Vamplew et al., 2022). In RL, this challenge is typically addressed by scalarizing the reward (*e.g.,* via a weighted linear combination) and training an agent to maximize the resulting aggregate signal. In practice, these weights are commonly chosen through trial-and-error, a nearly universal strategy among expert practitioners (Booth et al., 2023; Knox et al., 2023). Despite its prevalence, this approach remain largely underexamined. Manual weight tuning can overfit reward design to specific algorithms or hyperparameters, compromising fair evaluation and generalization (Booth et al., 2023). Furthermore, scalarization can be highly sensitive: small weight changes can yield drastically different behaviors upon retraining, making it difficult to obtain a stable or predictable trade-off among objectives.

Multi-objective reinforcement learning (MORL) involves simultaneous optimization of conflicting objectives. Unlike single-objective RL, MORL methods typically output a set of policies and their corresponding evaluations, referred to as the Pareto set (PS) and Pareto front (PF), respectively (Hayes et al., 2022; Felten, 2024). By exposing the full spectrum of possibly optimal returns, this approach facilitates a more transparent decision-making process, allowing the user to select a policy that aligns with their implicit preferences through an informed analysis of the trade-offs.

However, focusing exclusively on trade-offs in the objective space can obscure substantial behavioral discrepancies between policies. Since policy evaluations summarize performance, they can mask significant behavioral differences. This leads to scenarios in which qualitatively distinct behaviors appear indistinguishable because they yield identical or proximal points in the objective space (Osika et al., 2024). Thus, selecting a policy based solely on its position on the PF may lead decision-makers to overlook critical variations in execution. This creates a risk of selecting a policy that satisfies numerical benchmarks while exhibiting undesirable or unexpected behavioral patterns in practice.

**Motivating Example.** To illustrate why evaluating behavioral dynamics alongside objective trade-offs is essential, we introduce a modified version of the well-established Deep Sea Treasure (DST) benchmark (Vamplew et al., 2011; Felten et al., 2022). In the standard DST task, an agent controls a submarine in a grid world to balance treasure value against time-to-target. We propose a variation, termed *Left–Right DST*, which preserves these two objectives but distributes treasures across the left and right sectors of the map. This environment is intentionally designed such that neighboring policies on the Pareto front exhibit radically different behaviors, such as committing to strictly leftward or rightward trajectories (see fig. 1). In an

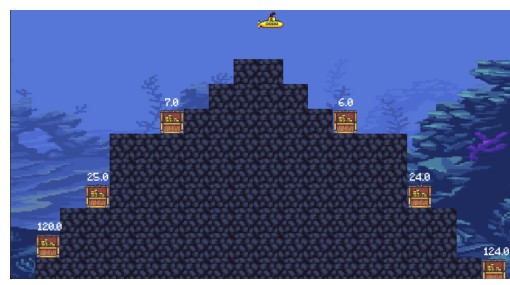

Figure 1: Left–Right DST.

ideal modeling scenario, these spatial preferences would be explicitly defined as additional objectives. However, in real-world applications, designers often lack the *a priori* information required to represent every behavioral nuance as a reward signal. Consider, for instance, a deployment where the submarine is a naval vessel: the right sector might represent international waters with higher latent risk, or the left might belong to a specific sovereign territory. While the distinction between "left" and "right" is transparent in our grid-world example, such behavioral divergences are often far more subtle in complex environments, and hard to explicitly model within the reward function. By using this straightforward case, we highlight a fundamental challenge: objective performance metrics can mask significant behavioral risks that only become apparent when analyzing *how* an agent achieves its goals.

In this work, we develop a diagnostic workflow for surfacing behavioral variation along the Pareto front that objective values alone do not reveal. Our workflow provides an automated means of flagging policy pairs whose behavioral differences are disproportionately large relative to their proximity in objective space, making them candidates for manual inspection. Practitioners can use these variations to detect whether the specified reward objectives are adequate for a given deployment or whether observed behavioral differences are harmful. However, judgment of those requires external information about stakeholder preferences, which is problem-specific. In particular, we:

- Identify the gap between objective-space proximity and behavioral proximity as a structural property of MORL policy sets that is not surfaced by current research, and provide illustrative examples of when this gap is and is not present.

- Introduce a modular workflow for constructing behavior-space representations and comparing their local structure to that of the Pareto front (see fig. 2 and section 3). The workflow is designed to accommodate alternative design choices, including different encoders, aggregation schemes, and metrics.

- Propose quantitative and visual diagnostics including trustworthiness, continuity, and behavior-objective scatter plots (described in section 3.3) to indicate where objective-space proximity is a poor summary of behavioral diversity, and to flag policy pairs warranting closer inspection.

The remainder of this paper is structured as follows. In section 2, we provide an overview of the current approaches on policy behavior characterization and behavior encoding and the background to our work. This is followed by a description of our workflow in section 3. In section 4, we demonstrate the applicability of the approach on a simple domain (DST) and show that it scales to more complex MuJoCo environments, indicating that the method remains effective as problem complexity increases. Finally, we include a consideration of the potential use and impact of this work in section 5.

## 2 Preliminaries

We begin by surveying related work. Then, we provide the necessary background on MORL and on encoders, which we leverage to construct trajectory-based behavioral representations.

## 2.1 Related Work

Prior work in multi-objective reinforcement learning mostly focused on learning an approximation of the PF in the objective space, evaluating for example through hypervolume (Van Moffaert & Nowé, 2014) or frontier approximation (Parisi et al., 2016), without analyzing the behavioral differences between policies along this front. A notable exception is the method introduced for explainable MORL by Osika et al. (2024), which represents behavior through salient state transitions identified via Q-values. These transitions are distilled into video-based highlights and behavioral matrices, which are then used to cluster policies by combining behavioral descriptors with objective values in a multi-objective setting. While effective when Q-values are accessible, this approach is highly sensitive to its design choices and is only in scenarios where Q-values can be obtained.

To analyze behavioral differences, we survey a series of generative sequence models, which have attracted interest of recent works. Several works relied on the use of the Transformer (Vaswani et al., 2017) architecture, often trained through contrastive learning, to learn motion priors for forecasting (Vivekanandan et al., 2025), or a spatial ranking of the states in the trajectories (Chang et al., 2023), discarding the structure of action sequences and thus losing key information about behavior, a limitation shared also by other representation learning approaches (Carroll et al., 2022). Ge et al. (2025) propose Variational Trajectory Embeddings, which considers the structure of action sequences, and employs a probabilistic skill extractor to yield a sequence of latent skill distributions given as input to a transformer-based VAE that computes the final embedding's posterior. However, the skill-extraction stage requires the number of skills to be determined beforehand. Further dependence on external annotation is shared also by other approaches in goal-conditioned learning (Ajay et al., 2022) . Mone et al. (2026) proposed a contrastive framework to learn a discriminative embedding space for behavioral clustering directly from raw state-action sequences. Each trajectory is represented by a L2-normalized `[CLS]` token (Devlin et al., 2019), which collects information from all the steps in the trajectory, encoding it in a single vector (Zou et al., 2024). This approach allows us to characterize differences across policies purely through their behavioral patterns, without external skill annotations or rewards. In principle, any encoder structure can be used as part of our proposed workflow, and can be substituted with ease.

Our work is also related to the broader literature on reward specification and the gap between specified and true objectives (Hadfield-Menell et al., 2017; Leike et al., 2018). That literature addresses the harder problem of recovering or aligning with stakeholder preferences that were never fully specified. We do not attempt to solve that problem. Our workflow operates purely on the geometry of the specified objective space and the learned behavior space; it can flag that behavioral variation exists that the specified objectives do not capture, but it cannot determine whether that variation is deployment-relevant without additional information about true stakeholder preferences.

## 2.2 Multi-Objective Reinforcement Learning

We formally model the problem as a multi-objective Markov decision process (MOMDP), defined by the tuple $\langle S, A, T, \gamma, \mu, \vec{R} \rangle$. Here, $S$ denotes the state space, $A$ the action space, $T : S \times A \times S \to [0, 1]$ the transition kernel, $\gamma \in [0, 1)$ the discount factor, and $\mu : S \to [0, 1]$ the distribution over initial states. The key distinction from a standard MDP lies in the vector-valued reward function $\vec{R} : S \times A \times S \to \mathbb{R}^d$, which returns immediate rewards for $d \geq 2$ objectives (Roijers et al., 2013; Hayes et al., 2022; Felten, 2024). An agent follows a policy $\pi : S \times A \to [0, 1]$ that defines a distribution over actions given states. The vector-valued return of a policy $\pi$ (also illustrated in fig. 2) is the discounted sum of rewards:

$$\vec{v}^\pi = \mathbb{E}\left[\sum_{k=0}^{\infty} \gamma^k \vec{r}_{k+1} \;\middle|\; \pi, \mu\right], \quad \text{where } \vec{r}_{k+1} = \vec{R}(s_k, a_k, s_{k+1}). \tag{1}$$

**Solution sets** Because the resulting value vector $\vec{v}^\pi \in \mathbb{R}^d$ only defines a partial order over policies, the concept of optimality in MORL relies on Pareto dominance $\succ_P$. One policy $\pi'$ dominates another policy $\pi$ if $\vec{v}^{\pi'}$ is non-inferior to $\vec{v}^\pi$ across all objectives and strictly superior in at least one objective. Formally, dominance is defined as: $\vec{v}^{\pi'} \succ_P \vec{v}^\pi \iff v_i^{\pi'} \geq v_i^\pi \forall i \in [1, d] \wedge \exists j \in [1, d] : v_j^{\pi'} > v_j^\pi$.

For a candidate policy set $\Pi$, the Pareto set comprises all policies that are not dominated by any other policy in $\Pi$: PS $= \left\{ \pi \in \Pi \mid \nexists \pi' \in \Pi \text{ such that } \vec{v}^{\pi'} \succ_P \vec{v}^{\pi} \right\}$. The PF is the associated set of return vectors, representing the boundary of the achievable returns in the objective space.

In practice, these solution sets are expensive to compute. Thus, algorithm designers usually convert the MOMDP into a simple MDP by scalarizing the reward vector, *e.g.,* using a weighted sum. The weights employed in such scalarization functions are often set *a priori* and adjusted through trial-and-error processes (Wurman et al., 2022; Booth et al., 2023). However, this widespread practice of trial-and-error reward design can be problematic. First, reward functions can be overfit to specific algorithms and hyperparameters, meaning that performance rankings across different reward functions are largely uncorrelated when the learning context changes (Booth et al., 2023). Second, small adjustments in scalarization weights can lead to drastic behavioral changes (Vamplew et al., 2022), particularly because most published reward functions exhibit near-universal design flaws, including unsafe reward shaping and risk tolerance misspecification (Knox et al., 2023). Thus, reward function design requires more principled approaches than ad-hoc manual tuning.

## 2.3 Encoder for Trajectory Representation

An encoder can be formalized as a parametrized mapping $E$, from an input space $\mathcal{X}$ to a latent space $\mathcal{Z}$, $E : \mathcal{X} \to \mathcal{Z}$. The aim of an encoder is to model the input data $x$ and transform it into a compact, high-level and meaningful embedding $z$, so that $z = E(x)$.

In our work, the encoder follows the architecture and training procedure of the Behavioral Encoder introduced by Mone et al. (2026). The goal of this encoder is to a obtain geometrically meaningful and robust representation of the policy behavior, given an input as sequences of state-action pairs. The training objective proposed by Mone et al. (2026) is a composite contrastive loss function, $\mathcal{L}_{\text{total}}$, a weighted linear combination of different components:

$$\mathcal{L}_{\text{total}} = \alpha \mathcal{L}_{\text{CLS}} + \beta \mathcal{L}_{\text{DIM}} + \gamma \mathcal{L}_{\text{SEG}} + \delta \mathcal{L}_{\text{PAIR}},$$

where $\alpha, \beta, \gamma$ and $\delta$ terms are non-negative scalar weights. The training procedure follows the contrastive strategy introduced by Gao et al. (2021), generating two dropout-augmented embeddings $z_{\tau_i}^1, z_{\tau_i}^2$ of the same input trajectory $\tau_i$ within a batch of size $N$ by passing $\tau_i$ twice to the model during training. The $\mathcal{L}_{\text{CLS}}, \mathcal{L}_{\text{SEG}}$ and $\mathcal{L}_{\text{PAIR}}$ components apply a symmetric InfoNCE loss (Oord et al., 2018; Chen et al., 2020) on different granularity levels of the input trajectory. The InfoNCE loss is then defined as:

$$\ell(z_{\tau_i}^1, z_{\tau_i}^2) = -\log \frac{\exp(\text{sim}(z_{\tau_i}^1, z_{\tau_i}^2)/\rho)}{\sum_{k=1, k \neq i}^{2N} \exp(\text{sim}(z_{\tau_i}^1, z_{\tau_k}^2)/\rho)},$$

with $\mathcal{L}_{\text{CLS}}$ applying it between complete trajectories, $\mathcal{L}_{\text{SEG}}$ applies it between a complete trajectory and the trajectory segments, and $\mathcal{L}_{\text{PAIR}}$ applies it between segments. The remaining loss component, $\mathcal{L}_{\text{DIM}}$, is a Deep InfoMax (DIM) (Hjelm et al., 2019) regularizer loss, which encourages $z_{\tau_i}$ to capture semantic information shared across all local token embeddings $\{t_k\}$ within the same trajectory.

## 2.4 Objective–Behavior Misalignment

We hereby provide a definition of the *objective-behavior misalignment*, formalizing the geometry mismatch between the objective space and the behavioral space.

**Definition 1 (Objective–Behavior misalignment)** *Let* PS $= \{\pi_1, \dots, \pi_N\}$ *be a Pareto Set with value vectors* $\{\vec{v}^{\pi_i}\} \subset \mathbb{R}^d$ *and aggregated behavioral embeddings* $z(\pi_i) \in \mathbb{R}^d$ *as defined in eq. (1) and Section 2.3 respectively.*

*Considering an* objective–behavior map *as* $\varphi : PF \to \mathcal{Z}, \quad \varphi(\vec{v}^{\pi_i}) = z(\pi_i)$, *the PS exhibits objective-behavior misalignment if* $\varphi$ *fails to preserve neighborhoods or have irregular distances between policies, when* $\exists (i, j)$ *such that* $j \in \mathcal{N}_{PF}^k(i) \setminus \mathcal{N}_{\mathcal{Z}}^k(i)$ *or* $j \in \mathcal{N}_{\mathcal{Z}}^k(i) \setminus \mathcal{N}_P^k F(i)$ *and there is high variance in the ratio* $\rho(i, j) = d_{\mathcal{Z}}(i, j)/d_{PF}(i, j)$ *across neighboring pairs* $(i, j)$ *in PF, with* $\mathcal{N}_{\mathcal{S}}^k(i)$ *the k nearest neighbors of i.*

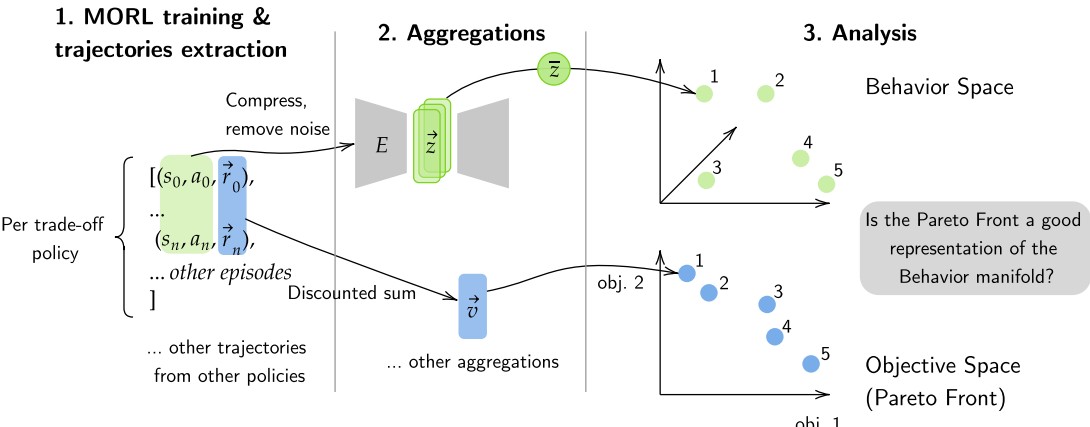

Figure 2: Overview of the proposed model-agnostic workflow. It extracts trajectories and expected returns from various Pareto optimal policies. Then, it compresses the trajectories to describe them through their patterns, keeping important features. Finally, it compares the two spaces to detect inconsistencies when moving from one policy to another in the behavior and the objective spaces.

Definition 1 formalizes the phenomenon our workflow detects: pairs of policies whose relationship in objective space is a poor predictor of their relationship in behavior space. This definition is a purely geometric property of $\varphi$: it flags pairs warranting inspection, but it does not, on its own, determine whether a flagged difference is deployment-relevant. That judgment requires external information about stakeholder preferences, and our workflow deliberately delegates it to the analyst rather than resolving it automatically.

## 3 A Method for Behavior-Objective Space Analysis

We propose a three-step diagnostic workflow to surface behavioral variation along the Pareto front that objective values alone do not reveal, as illustrated in fig. 2. Concretely, the workflow identifies policy pairs whose behaviors differ substantially despite their proximity in objective space: pairs that a practitioner relying only on the Pareto front would have no reason to inspect further. The framework is intentionally modular, facilitating extensions such as utilizing different trajectory encoders, integrating with various MORL algorithms, or employing alternative evaluation metrics.

### 3.1 MORL Training and Trajectories Extraction

We first train a MORL agent to obtain a set of policies representing diverse trade-offs, thereby defining the PS and the PF. Our analysis process is agnostic to how policies are obtained: for grid domains (*e.g.,* DST), the optimal policies are computed analytically; for the more complex environments, we train our policy set using MORL based on decomposition (MORL/D) (Felten et al., 2024). MORL/D follows an evolutionary approach that maintains a population of Pareto dominant policies, each trained with respect to a distinct weight vector utilized in a linear scalarization of the reward vector. We note that, while more sample-efficient variants exist, these were deliberately omitted to adhere closely to the common trial-and-error process used for defining the weights applied to the scalarization function in standard single-objective RL contexts. For each policy in the PS, we extract 50 (user-defined parameter) trajectories, which serve as input to the encoding step that is described next, together with their vector-valued returns. In the more complex domains, these raw trajectories are high-dimensional and may contain substantial noise.

### 3.2 Aggregation via Behavior Encoding

In the second step, we process the extracted trajectories to obtain meaningful representations of policy behavior. Following Mone et al. (2026), we use a learned trajectory encoder, $E$, to generate embeddings, $\vec{z}$,

trained contrastively to summarize the behavioral patterns of individual rollouts, and map similar ones close together in the space. We include a small variance–covariance regularizer to avoid degenerate embeddings and make full use of the available embedding dimensions, as in Bardes et al. (2022), resulting in the following equation: $L_{VC} = \lambda_{\text{var}} \sum_j \max(0, 1 - \sigma_j) + \lambda_{\text{cov}} \sum_{i \neq j} \text{Cov}(z)_{ij}^2$. The goal is to ensure a well-structured geometry for the space reflecting the behavioral characteristics of the trajectories.

We aggregate the embeddings to obtain a single vector $\bar{z}$ per policy, which defines the behavior manifold $\mathcal{Z}$. Formally, for a policy $\pi$, the aggregated behavioral embedding is defined as: $\bar{z}(\pi) = \text{agg}_{\tau_{1:K}}(E(\tau^\pi)) \in \mathbb{R}^d$, where $\tau_{1:K}^\pi$ are $K$ rollouts of state-action pairs of $\pi$, and agg is an aggregating function (in our experiments, we aggregate them by the mean). This representation captures the *behavior* induced by the policies rather than how they *score*, providing a basis for comparing objective and behavioral spaces and analyzing their alignment. The objective aggregations are drawn from the definition of $\vec{v}^\pi$, eq. (1).

### 3.3 Analysis

In the third step, we compare the derived behavior manifold $\mathcal{Z}$ with the PF. The central question guiding this analysis is: "Are policies that are neighboring in the objective space also behaviorally similar, and if not, which pairs differ most?" Low values of the metrics we introduce below indicate that the objective space is a poor summary of behavioral diversity: a decision-maker relying only on Pareto-front proximity to assess policy similarity would be misled. The practical implication is that practitioners who have any reason to care about behavioral differences–whether or not those differences are captured by the reward function–should not rely on the Pareto front alone for policy selection. We adapt concepts from other disciplines, such as nonlinear projection analysis and optimization landscape analysis, for measuring objective-behavior discrepancies in (MO)RL. Two metrics are drawn from nonlinear projection analysis, originally developed for dimensionality reduction, which we adapt to quantify how well behavioral differences preserve objective space structure. We also adapt a scatterplot analysis based on Lipschitz continuity from optimization landscape analysis to assess the local sensitivity of behaviors to objective changes. Although these methods were designed for different purposes, we demonstrate their effectiveness when adapted to our application.

**Trustworthiness and continuity** Let $N$ be the number of data points, $\mathcal{N}_{\mathcal{S}}^k(i)$ the set of $k$ nearest neighbors of policy $i$ in space $\mathcal{S} \in \{\text{PF}, \mathcal{Z}\}$, and $r_{\mathcal{S}}(i,j)$ denote the rank of policy $j$ with respect to policy $i$ in space $\mathcal{S}$. We define a generic neighborhood preservation metric for a mapping from space $\mathcal{S}_1$ to space $\mathcal{S}_2$ as

$$Q_k(\mathcal{S}_1 \to \mathcal{S}_2) = 1 - \frac{2}{Nk(2N - 3k - 1)} \sum_{i=1}^{N} \sum_{j \in \mathcal{N}_{\mathcal{S}_2}^k(i) \backslash \mathcal{N}_{\mathcal{S}_1}^k(i)} \left( r_{\mathcal{S}_1}(i,j) - k \right).$$

Trustworthiness $T(k)$ and continuity $C(k)$ (Venna & Kaski, 2001) correspond to opposite directions of neighborhood preservation between the Pareto front and the behavior manifold:

$$T(k) = Q_k(\text{PF} \to \mathcal{Z}), \qquad C(k) = Q_k(\mathcal{Z} \to \text{PF}).$$

Together, these metrics provide complementary insights into the embedding quality: high values (close to 1) for both metrics indicate that the objective-behavior mapping preserves local geometry without introducing spurious neighbors or losing existing relationships. We report trustworthiness and continuity using $k = 2$, which probes the most local neighborhood structure: the regime most relevant to our diagnostics, since misalignment concerns policies that are immediate neighbors on the Pareto front. A $k$-sensitivity analysis on the tested environments confirms that this choice lies well within the stable range of both metrics; the full analysis is provided in Appendix C.2.

**Lipschitz scatterplots** Since the above metrics are based on neighborhood rank orderings, they provide only a generalized view of local structure preservation and may not capture the magnitude of distance changes. To address this, we introduce scatterplots inspired by the concept of Lipschitz continuity (Cobzaş et al., 2019). A function $f$ is Lipschitz continuous

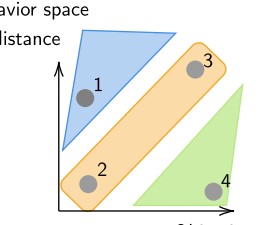

if its rate of variation is bounded: $|f(x_i) - f(x_{i+1})| \leq L |x_i - x_{i+1}|$. Intuitively, this concept places an upper bound on how quickly behaviors can change relative to their objective value when moving from policy $i$ to policy $i + 1$ in the PF. To order the policies, we start from one end of the PF and take the nearest non-selected neighbor until all policies have been selected; this gives lexicographic ordering in 2-dimensional objective space, and a decent, although not perfect, heuristic in more than two dimensions.

We utilize this concept to analyze the relationship between distances in the objective and behavior spaces. We construct a scatter plot where each point represents a pair of neighboring policies in the objective space, see fig. 3. The $x$-axis displays their distance in the objective space ($|f(x_i) - f(x_{i+1})|$), and the $y$-axis shows the corresponding distance in the behavior space ($|x_i - x_{i+1}|$).

We interpret the resulting plots as follows. **Diagonal points** (points 2 and 3 in fig. 3): Represent smooth transitions where behavioral changes correspond predictably to objective changes. **Lower-right region** (point 4): Indicates that the objective difference is unusually high relative to the behavioral difference, we noticed it often corresponds to "holes" or sparse regions in the PF. **Upper-left region** (point 1): Policies whose behavioral differences are significantly larger than their objective differences. These points flag policy pairs that are close on the Pareto front but behaviorally distinct: pairs a practitioner would have no reason to inspect without a tool like ours. The two regions reflect the conditions of Definition 1 causing objective-behavior misalignment, with the upper-left region having large $\rho(i, j)$, and the lower-right region having small $\rho(i, j)$. Whether the behavioral difference is deployment-relevant depends on the application; our workflow surfaces these pairs for human judgment rather than resolving that question automatically. We refer to this as the *critical region*.

**Eyeball test** Since a behavior is inherently difficult to characterize using purely quantitative measures, we complement the numerical analysis with qualitative inspection. Specifically, we render and examine policy rollouts to directly observe the unfolding of trajectories. For this purpose, we visualize five episodes per policy and compare the resulting behaviors, especially for policies identified as distinct through the numerical analysis. This allows us to assess whether the patterns and distinctions identified by the tools explained above are also reflected in the raw trajectories.

## 4 Results

First, we present the results for grid environments based on the DST. Specifically, we verify whether our encoder captures trends similar to human-designed embeddings, and analyze how policies' behavior, represented by the transformer embeddings (mean and standard deviation over five random seeds), relates to the Pareto front across different DST instances. Then, we extend our analyses to continuous environments with 2 and 3 objectives, MO-HalfCheetah and MO-Hopper (Todorov et al., 2012; Alegre et al., 2022; Felten et al., 2023), to evaluate the scalability of our findings. A single MORL/D run produces the policies for these environments, as multiple runs would find different policies and therefore different PF/PS. The trajectories produces by these policies are given in input to the encoder, and we report the mean and standard deviation over five random seeds. As different encoding approaches are possible, our approach is modular enough to allow for substitutions, as we show with our ablations. We performed an ablation study considering two simpler encoder models, a simple MLP encoder and an LSTM-based one. These encoders were able to produce correct results in terms of Trustworthiness and Continuity. Nonetheless, the encoder proposed by Mone et al. (2026) showed to be more robust and reliable in the scatterplot distances, validating our architectural choice for the encoder. For the hyperparameters, we deployed the method with the suggested configuration (further explained in the Appendix A.2), with the only exception of the latent dimension, which we fix at 3 for all the environments. The rationale behind this decision is to avoid the curse of dimensionality, and to enforce the distances to be meaningful by construction.

The results of the baseline model used for the ablation are reported in Appendix C.

### 4.1 Experimental Setup

To generate our sets of policies, we use the MORL/D implementation from MORL-Baselines (Felten et al., 2023). The hyperparameters for MORL/D and for the encoder training are listed in section A.2. We note that the workflow scales linearly in the number of policies (each policy contributes a fixed number of roll-outs encoded independently), and the encoder itself is lightweight: in our experiments, it was trained on a consumer laptop (MacBook Pro M1). We evaluate our method on two new variants of the DST benchmark (Vamplew et al., 2011; Felten et al., 2022). *Smooth DST* (fig. 8) places treasures along a descending curve, yielding a smooth PF in which objective values and behaviors are fully aligned. In contrast, *Left–Right DST* (fig. 1) places treasures on both sides of the agent's starting position, yielding multiple distinct trajectories with similar returns. This creates a misalignment between behavioral and objective spaces and exposes the limitations of relying solely on the PF to choose a policy (see Appendix B for more details). Due to the simplicity of DST, we construct *manual behavior embeddings*, defined as $(-\#\text{left} + \#\text{right}, \#\text{down})$, where $\#$ denotes action frequency, to interpret and compare with our learned transformer embeddings.

### 4.2 Deep Sea Treasure

As shown in fig. 4, policies that are close in objective space can differ substantially in behavior space. In the Left–Right DST, embeddings cluster clearly according to whether the agent navigates left or right, a distinction invisible in the objective space. For Smooth DST (fig. 5), no such clustering is expected, and both embeddings and objective values place similar policies close to each other, reflecting a smooth and aligned mapping. In both settings, transformer embeddings recover the same behavioral structure as the manually designed ones (middle panel vs. left one). A more detailed comparison between learned embeddings and manual embeddings can be found in Appendix C.

When quantitatively comparing spaces in table 1, trustworthiness and continuity are low for Left–Right DST (around 0.5), confirming that the objective space fails to preserve behavioral neighborhoods in the misaligned setting. In contrast, both metrics are near 1 for Smooth DST, indicating a faithful mapping between objectives and behavior. In both settings, the values for PF vs. transformer embeddings closely match those for PF vs. manual embeddings, further validating that our transformer encoder captures the same behavioral relationships as the manually designed embeddings.

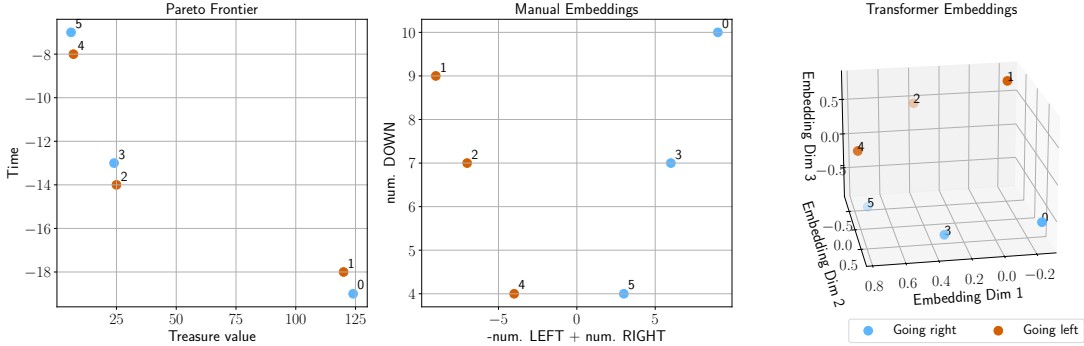

Figure 4: Pareto front and behavioral embeddings for Left–Right DST. Colours indicate directional behavior (left vs. right). Both manual and transformer embeddings produce consistent behavioral clustering.

Another component of our assessment is a qualitative analysis of local relationships between policies using the Lipschitz-inspired scatter plots. In Left–Right DST, the largest discrepancies occur between policies 0 and 1: they are very close in the objective space but far apart in the behavior space (Figure 6a). Conversely, policies 1 and 2 are farthest apart in the objective space but closest in the behavior space, reflected as points in the bottom-right of the scatter plot. For Smooth DST (Figure 6b), consecutive policies are equidistant in both objective and behavioral space. In principle, the scatter plot should collapse to a single point for Smooth DST. However, the encoder introduces moderate variation in behavior-space distances (range 0.2–0.55), likely due to its approximate nature. Although this variation warrants further investigation, its magnitude remains

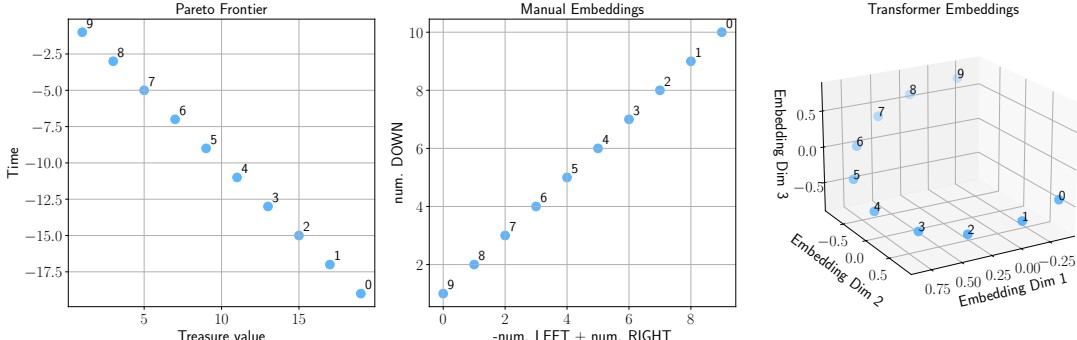

Figure 5: Pareto front and behavioral embeddings for Smooth DST. Both manual and transformer embeddings produce consistent behavioral clustering.

Table 1: Global metrics for objective-space versus behavior-space shift detection.

| Env. | Comparison | Trustworth. | Continuity | Exp. score |
|---|---|---|---|---|
| Left–Right | PF vs. Manual embed. | $0.57 \pm 0.00$ | $0.50 \pm 0.00$ | low |
|  | PF vs. Transformer embed. | $0.51 \pm 0.07$ | $0.47 \pm 0.04$ | low |
| Smooth | PF vs. Manual embed. | $1.00 \pm 0.00$ | $1.00 \pm 0.00$ | high |
|  | PF vs. Transformer embed. | $0.99 \pm 0.01$ | $1.00 \pm 0.00$ | high |

substantially smaller than the behavioral differences observed in Left–Right DST. Overall, these results indicate that our transformer-based encoder is well suited for analyzing behavior–objective relationships.

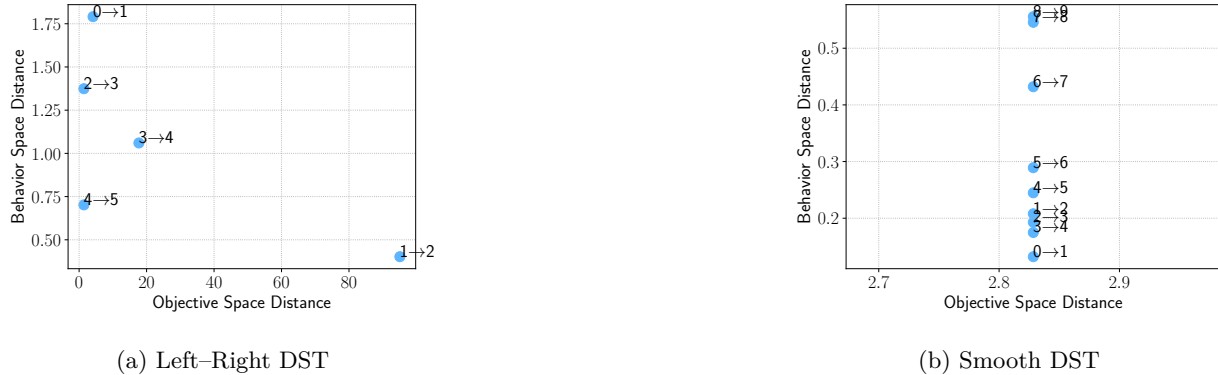

(a) Left–Right DST

(b) Smooth DST

Figure 6: Distances between consecutive policies in the objective and behavior space (mean over random seeds 0–4). Please note that the axes have different scales in the two plots.

### 4.3 MuJoCo environments

Having demonstrated the effectiveness of our approach on DST, we extend the analysis to more complex environments: 2-objective MO-HalfCheetah and 3-objective MO-Hopper. As shown in table 2, trustworthiness and continuity are close to 1 for both environments, indicating strong global alignment between the objective and behavior spaces on average. MO-HalfCheetah achieves slightly higher values, suggesting marginally stronger alignment than MO-Hopper.

Next, we analyze the scatter plots in fig. 7. The PS contains 80 policies for Cheetah and 160 policies for Hopper. Examining the scatter plots (figs. 7b and 7d), we observe that the majority of points lie in the

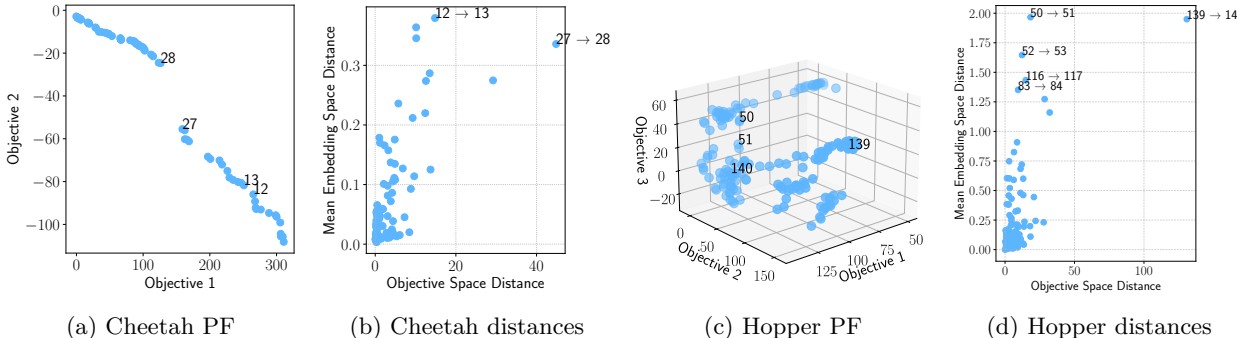

(a) Cheetah PF      (b) Cheetah distances      (c) Hopper PF      (d) Hopper distances

Figure 7: Pareto fronts and mean transformer embedding distances between consecutive policies in the objective and behavior space for MO-HalfCheetah and MO-Hopper. Highlighted policy pairs occupy either the critical upper-left region (close in objective space, far in behavior space) or exhibit large distances in both spaces, and are selected for further trajectory analysis. Please note that the axes have different scales in the distance plots.

lower-left region. This means, in general, policies have similar (small) distances to each other in behavior and the objective space. This observation is consistent with the global metrics.

However, an important difference emerges when considering the range of distances in the embedding space. For Cheetah, the distance range is relatively small (0–0.3), whereas for Hopper it is substantially larger (0–2). For comparison, in the Left–Right DST setting the embedding-space distances range from 0.5 to 1.75, while in the Smooth DST case they range from 0.2 to 0.5. This suggests that the magnitude of distances plays an important role and that scatter

Table 2: Trustworthiness and continuity of PF vs. learned embeddings.

| Environment | Trust. | Cont. |
|---|---|---|
| MO-HalfCheetah | $0.991 \pm 0.001$ | $0.991 \pm 0.001$ |
| MO-Hopper | $0.945 \pm 0.008$ | $0.951 \pm 0.004$ |

plots should be interpreted not only visually but also in conjunction with the numerical spread of distances. To investigate this further, we focus on two edge-case policy pairs from each environment in the following analysis. Specifically, we select policies 12–13 for MO-HalfCheetah and policies 50–51 for MO-Hopper. Detailed policy-by-policy breakdowns are provided in Appendix D. We render the selected policy pairs for five episodes each and assess whether the differences indicated in the scatter plot are perceptible to a human observer. The rendered videos, together with additional analysis, are provided in the accompanying interactive version of the paper[1] and in the Table 9 in the Appendix D.

For MO-HalfCheetah, the behavioral differences between the selected policy pairs are subtle and not immediately apparent to the human eye, possibly related to the relatively low range of mean behavior space distances (0–0.4, see fig. 7b), analogous to what was observed in Smooth DST. In contrast, for MO-Hopper, the differences are clearly visible: policy 50 consistently lunges forward with its torso using large-magnitude actions, while policy 51 maintains a more upright posture with smaller, smoother actions. Since the behavior space is constructed from trajectories comprising both actions and observations (including joint angles), this distinction is faithfully captured by the learned embeddings. This pattern holds consistently across other policy pairs in the critical region (52–53, 116–117, 83–84), where larger, more abrupt actions produce forward lunges or higher jumps at the cost of stability, while smoother actions yield more controlled locomotion.

## 5 Conclusion

We propose an exploratory diagnostic workflow for inspecting behavioral variation along the Pareto front in MORL. The workflow surfaces policy pairs whose behaviors differ substantially despite their proximity in objective space, a variation that standard Pareto front analysis does not reveal. We do not claim to determine whether the specified reward objectives are adequate for deployment, nor whether observed behavioral

---

[1]https://magnificent-marzipan-25536f.netlify.app/

differences are harmful; those judgments require external information about stakeholder preferences that our workflow does not provide. What it does provide is an automated, scalable means of flagging candidates for manual inspection, complementing the numerical summaries that MORL algorithms typically present to decision-makers. By applying both quantitative and qualitative diagnostics across grid worlds and continuous domains, we demonstrate that the approach is effective across a range of problem complexities.

The proposed framework opens several avenues for application:

**Detecting behavioral instability in single-objective RL:** In single-objective RL, our scatter plots can identify reward functions prone to instability. Policies residing in "critical regions" indicate that marginal changes in reward structure may trigger disproportionate behavioral shifts, alerting practitioners to potential sensitivities in reward design.

**Decision support and behavioral analysis in MORL:** Our work assesses whether objective-space proximity reliably predicts behavioral similarity. Low trustworthiness or continuity scores, visualized through local scatter plots, alert decision-makers to "critical regions" where neighboring policies exhibit divergent behaviors, necessitating manual inspection. The proposed approach also enables a systematic analysis of behavioral diversity analogous to analyzing decision spaces in traditional optimization (Osika et al., 2023; Bandaru et al., 2017), providing a more transparent foundation for policy selection in complex MORL applications.

**Surfacing behaviorally distinct policies for inspection.** When behavioral analysis reveals that policies achieving similar objective values employ substantially different strategies, our workflow flags these pairs for manual inspection. Whether the observed behavioral differences indicate an incomplete problem formulation–for instance, a reward function that fails to capture all deployment-relevant features–cannot be determined from the workflow alone, as this requires external information about true stakeholder preferences. However, by making such variation visible, the workflow gives practitioners the opportunity to assess whether their problem formulation is adequate, and to refine it if not. This is particularly valuable because, as shown in prior work (Booth et al., 2023), practitioners often define reward functions without anticipating all behaviorally relevant dimensions of the task.

*An illustrative scenario.* Consider a deployment-oriented reading of the Left–Right DST environment. Suppose the submarine operates in a conflict setting where the map is divided into zones: certain regions (e.g., the left sector) cannot be traversed due to restricted or hostile territory. Ideally, this constraint would be encoded as an additional objective, but in practice the restriction may not have been known at design time or may simply have been overlooked, a failure mode that prior work shows is common even among expert practitioners (Booth et al., 2023). In this case, the Pareto front alone gives the decision-maker no indication that some policies traverse the restricted zone while others do not. Our workflow flags exactly these policy pairs: nearly identical in objective space, drastically different in behavior. Upon inspection, the practitioner realizes that a deployment-critical feature was omitted from the problem formulation and can revise the objectives accordingly. While in the DST setting this challenge may appear artificial as the environment is a simple synthetic problem designed to make our approach easy to visualize, and the omitted constraint would be trivial to encode once detected, the situation compounds in continuous domains. In MO-Hopper, a practitioner may not know in advance that movement style (e.g., abrupt, large-magnitude actions versus smooth, controlled locomotion) matters for their deployment, and even when aware of it, such behavioral properties are genuinely difficult to formalize as reward terms. Observing behaviorally distinct policies that achieve similar returns helps the practitioner understand the problem itself better: it reveals dimensions of behavioral variation they had not anticipated, and thereby informs how the objectives should be re-defined. This reflects a broader reality of applied (MO)RL: there is exists uncertainty about problem formulation: which objectives matter, and how they should be modeled, is rarely known fully a priori. Our diagnostics support this iterative refinement loop by making visible the behavioral variation that the current formulation does not capture.

**Algorithm benchmarking:** Our work introduces a dimension for benchmarking that extends beyond standard convergence and diversity measures, such as hypervolume. This enables the evaluation of MORL algorithms based on the behavioral stability and predictability of the frontiers they generate, providing a more holistic assessment of an algorithm's reliability in practical deployments.

While this workflow represents a significant step toward understanding the objective-behavior relationship, it opens several avenues for future refinement. First, it is important to emphasize that our workflow is modular and could be instantiated with different components, *i.e.,* encoder, aggregation function, and metrics. Furthermore, our methodology relies on a specific policy ordering heuristic that, while effective for two objectives, becomes increasingly complex in higher-dimensional objective spaces (*e.g.,* the discontinuities (139–140) observed in fig. 7c). Future research into robust policy sequencing could enhance the reliability of these diagnostics. Additionally, the fidelity of our analysis is sensitive to the quality of the learned behavioral embedding, which necessitates hyperparameter tuning for the encoder that we performed manually. For instance, we constrained the latent space to 3 dimensions to maintain interpretability and avoid nested dimensionality reduction. Future work could explore automated dimensionality selection (Chen & Fuge, 2024) to dynamically capture the intrinsic complexity of different behavioral domains.

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

## A   Reproducibility

### A.1   Code

Our code and data are publicly available at https://github.com/ffelten/Behavior-vs-Objective-Space

### A.2   Hyperparameters

Table 3 lists the hyperparameters used for training the MORL/D policies. Policy optimization within MORL/D relies on the soft actor-critic (SAC) algorithm (Haarnoja et al., 2018). The rest of the hyperparameters have been set to the default values in MORL-Baselines. table 4 lists the hyperparameters used for the training of the Behavioral Encoder. For the encoder, we use the hyperparameters suggested by the authors of the original work, with the exception of the embedding dimension (emb_dim) which we fix to 3 to avoid the curse of dimensionality and enforce meaningful distances by construction. The maximum length (max_len) simply represents the maximum length of the trajectory between states and actions.

### A.3   Hardware

We trained our MORL policies on the ETH Zurich's high performance computer, which is equipped with NVIDIA RTX 4090 GPUs. The Encoder has been trained on a MacBook Pro 2020 with M1 Apple Silicon chip.

## B   Environments

**Deep Sea Treasure environment**   We evaluate our approach using the Deep Sea Treasure (DST) environment, a classic benchmark in multi-objective reinforcement learning. In DST, the agent controls a submarine navigating a 2D grid world with discrete coordinates [0,10] along both axes. The action space is also discrete, allowing the agent to move *up* (0), *down* (1), *left* (2), or *right* (3). Each episode terminates when the submarine reaches a treasure, with rewards determined by the treasure's position and time taken to reach it.

We use two variants of the environment: Smooth and Left–Right, each designed to illustrate different relationships between the objective and behavior spaces.

Table 3: Hyperparameters used for training MORL/D.

| Hyperparameter | Value |
| --- | --- |
| Number of timesteps | 1,000,000 |
| $\gamma$ | 0.99 |
| Reference point | $[-100, -100]$ for Cheetah, $[-100, -100, -100]$ for Hopper |
| Random seed | 0 |
| Scalarization method | Weighted Sum |
| Evaluation mode | SER |
| Policy architecture | MOSAC |
| Shared replay buffer | False |
| Weight adaptation method | None |
| Exchange frequency | 10,000 |

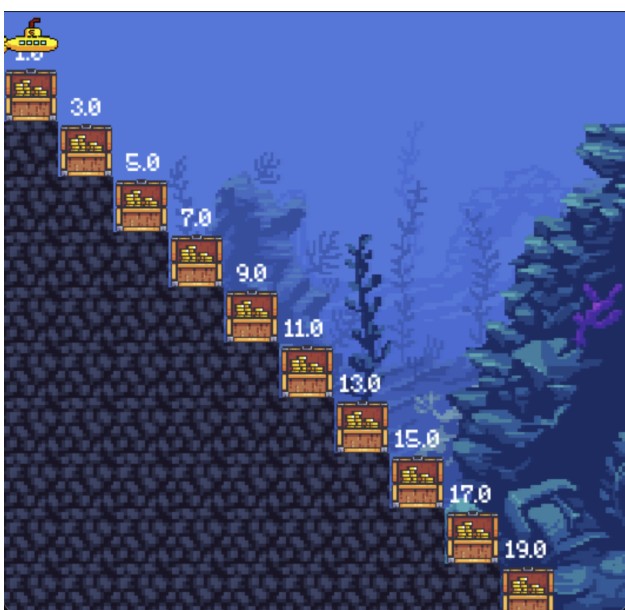

Figure 8: Smooth DST.

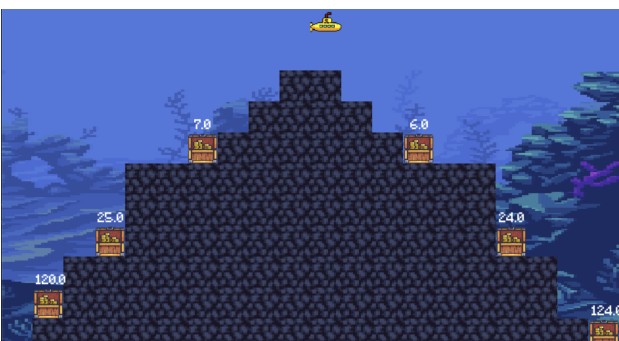

Figure 9: Left–Right DST.

Table 4: Hyperparameters used for the Behavioral Encoder training.

| Parameter | Value | Notes |
|---|---|---|
| **Training** | | |
| epochs | 100 | – |
| batch_size | 32 | – |
| lr | 3e−4 | AdamW |
| seed | 0 | – |
| **Encoder** | | |
| emb_dim | 3 | CLS embedding dimension |
| nheads | 1 | transformer attention heads |
| nlayers | 2 | transformer layers |
| d_hid | $32 - 128$ | hidden dim (DST 32, MuJoCo 128) |
| dropout | 0.1 | dropout probability |
| max_len | 20-100 | encoder max trajectory length (20 for DST, 100 MuJoCo) |
| gaussian_m_state | 128 | RFF size |
| gaussian_sigma_state | $0.01 - 0.001$ | RFF sigma (DST 0.01, MuJoCo 0.001) |
| gaussian_m_action | 32 | RFF size for actions |
| gaussian_sigma_action | $0.01 - 0.001$ | RFF sigma for actions |
| **Loss / regularizers** | | |
| $\alpha$ | 1.0 | $\mathcal{L}_{\text{CLS}}$ weight |
| $\beta$ | 1.0 | $\mathcal{L}_{\text{DIM}}$ weight |
| $\gamma$ | 1.0 | $\mathcal{L}_{\text{SEG}}$ weight |
| $\delta$ | 1.0 | $\mathcal{L}_{\text{PAIR}}$ weight |
| vc_weight | 0.05 | VC regularizer weight |
| temperature | 0.05 | InfoNCE temperature |
| VC std_coeff | 25.0 | Variance term coefficient |
| VC cov_coeff | 1.0 | Covariance term coefficient |

- **Smooth DST.** In this variant of the environment, the agent starts from the top-left position and moves to the right and down to collect treasures arranged along a descending curve (fig. 8). The objective and behavior spaces are fully aligned: progressing further along this path yields proportionally higher treasure rewards but also higher time costs. The name comes from the shape of the Pareto front, which is smooth (see Figure 5).

- **Left–Right DST.** In this variant, the agent may move either left or right to collect treasures that provide similar returns (fig. 9). This creates multiple distinct behaviors that achieve comparable objective values, introducing a natural misalignment between behavioral and objective spaces. This setting enables the study of scenarios where return-based metrics fail to reflect behavioral diversity.

**Half-Cheetah Two-Objective** Two-objective Half-Cheetah environment is a multi-objective extension of Gymnasium's `HalfCheetahEnv`. The agent controls a 2D cheetah with nine body parts and eight joints, applying torques to move forward efficiently.

The observation space consists of joint positions and velocities (17–18 dimensions, depending on whether the global x-coordinate is included). The reward function is two-dimensional:

1. Forward velocity (encouraging speed)

2. Control cost (penalizing large torques)

**MO-Hopper** The environment extends classical control setups by increasing the number of independent state and action variables. It features a 2D one-legged hopper composed of four linked components: a torso,

Table 5: Performance match between manual and transformer embeddings.

| Env. | Comparison | Trustworthiness | Continuity |
|------|-----------|-----------------|-----------|
| Left–Right | ME vs. TE | $0.98 \pm 0.03$ | $0.98 \pm 0.03$ |
| Smooth | ME vs. TE | $0.99 \pm 0.01$ | $1.00 \pm 0.00$ |

thigh, leg, and a single foot that supports the body. The agent moves forward by applying torques to the three joints connecting these segments, enabling hopping motions in the rightward direction.

The reward is three-dimensional:

1. Forward progress along the x-axis,

2. Vertical jump height along the z-axis,

3. Control cost of the applied actions.

## C   Additional results

### C.1   Manual embeddings vs. Transformer embeddings for DST

For DST, our experiments are structured in three stages.

**Is our encoder learning something close to what humans would do?** We first evaluate the transformer embeddings (TE) by comparing them to the manual embeddings (ME) both in aligned (Smooth DST) and misaligned (Left–Right DST) settings. Ideal alignment corresponds to metric values close to 1, indicating that the embeddings closely match each others. As shown in table 5, both trustworthiness and continuity are indeed near one.

This correspondence is also visually obvious. In Left–Right DST (fig. 4), MEs cluster clearly based on whether the agent moves left or right (not aligned with the PF). The TEs exhibit the same clustering pattern, confirming that the encoder captures behaviorally meaningful distinctions. For Smooth DST (fig. 5), distinct clusters are not expected, and we anticipate a smooth mapping from objectives to behavior. In this case, both MEs and the Pareto front place similar policies close to each other, a pattern that is preserved in the TEs.

**Is the PF a good representation of our manual embeddings?** Next, we establish baseline metrics by comparing the MEs with the corresponding objective vectors.

As shown in table 1, trustworthiness and continuity for PF vs. ME are low for the Left–Right DST, indicating poor neighborhood preservation between the spaces. This aligns with our expectations and confirms that the objective space does not reflect the behavioral distinctions in this misaligned setting. In contrast, for Smooth DST, policy neighborhoods are preserved between the manual embeddings and the Pareto front, with both metrics equal to 1 (for PF vs. ME case). This demonstrates a perfect mapping, consistent with the expected smooth relationship between objectives and behavior. These metrics establish the baseline for evaluating the TEs in further analyses.

Another component of our assessment is a qualitative analysis of local relationships between policies using the Lipschitz-inspired scatter plots. The Left–Right DST contains the perfect example of each case. In (fig. 4), the largest discrepancies occur between policies 0 and 1: they are very close in the objective space but far apart in the behavior space, as observed in the Pareto front and manual embeddings (Figure 10a). Conversely, policies 1 and 2 are farthest apart in the objective space but closest in the behavior space, reflected as points in the bottom-right of the scatter plot (see Figure 4). For Smooth DST (Figure 10c), all points collapse into a single location because the relative distances between policies are preserved across the objective and behavior spaces. This aligns with the constant, smooth spread observed in the Pareto front and manual embeddings (Figure 5).

**Is the PF a good representation of our learned embeddings?** Finally, we perform our main analysis by comparing the TEs with the objective vectors. This evaluation tests whether the TEs captures the same relationships as the MEs, accurately reflecting alignment in the Smooth DST environment while detecting divergence in the Left–Right DST.

Looking at global metrics such as trustworthiness and continuity, the PF vs. TE values closely match those of PF versus MEs: around 0.5 for Left–Right DST and near 1 for Smooth DST. This demonstrates that the TEs preserve the key structural relationships.

Qualitative comparisons of the scatter plots further support this conclusion. For Left–Right DST (Figure 10b), the relative relationships between policies are nearly identical to those observed in the manual embeddings. For Smooth DST (Figure 10d), variations in the behavior space (distance range 0.2–0.55) are observed, likely due to the nature of the encoder. While it does not seem to affect the global metrics, this point should be further studied in future work. Overall, these results confirm that our transformer-based encoder approach can be fit for studying behavior-objective relationships.

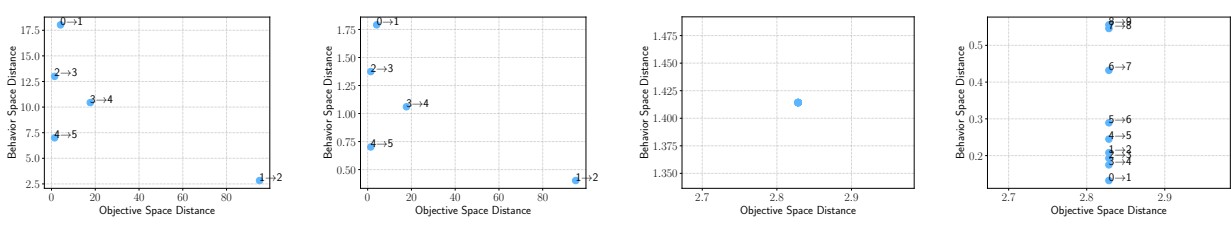

(a) Manual embeddings distances (Left–Right)  (b) Mean transformer distances (Left–Right)  (c) Manual embeddings distances (Smooth)  (d) Mean transformer distances (Smooth)

Figure 10: Distances between consecutive policies over the PF in the objective and behavior space (mean over different random seeds, 0–4) for Left–Right and Smooth DST.

## C.2 Sensitivity analysis for neighbourhood size

Trustworthiness and continuity are computed with respect to a neighborhood size $k$, which determines the locality of the analysis: small $k$ probes whether immediate neighbors are preserved between the objective and behavior spaces, while larger $k$ evaluates preservation of progressively coarser structure. To assess the sensitivity of our conclusions to this choice, we computed both metrics across the full range of admissible $k$ for MO-HalfCheetah and MO-Hopper (figs. 11 and 12). We restrict the sweep to these two environments, as the DST variants contain too few policies for a meaningful analysis: with $N$ policies, the metrics are well-defined only for $k < (2N - 1)/3$, leaving only a handful of admissible values in the DST settings.

Both metrics are stable across small-to-moderate neighborhood sizes: the regime relevant for our diagnostics, since misalignment is a local phenomenon. For MO-HalfCheetah, values remain above 0.95 up to $k \approx 25$ (roughly 30% of the 80 policies), and for MO-Hopper above 0.8 up to $k \approx 50$ (roughly 30% of the 160 policies). The gradual decline at moderate $k$ is expected: larger neighborhoods test preservation of increasingly global structure, which is naturally harder to maintain and is not the regime relevant to our local diagnostics. The accelerating drop as $k$ approaches $(2N - 1)/3$ is a normalization artifact: the factor $2N - 3k - 1$ in the denominator of $Q_k$ tends to zero, amplifying the penalty of each remaining rank violation, until the metric leaves its valid range entirely, explaining the collapse near $k \approx 53$ for MO-HalfCheetah ($N = 80$) and $k \approx 106$ for MO-Hopper ($N = 160$). Our reported results use $k = 2$, which lies well within the stable regime; the conclusions are unchanged for any reasonable choice of $k$.

## C.3 Ablation study results with the baseline network

As baseline, we provide an LSTM-based encoder (Section C.4) and a simple MLP encoder. The MLP encoder that flattens the entire trajectory into a single vector and maps it to one embedding $z$. Per-frame observations and actions are concatenated per step, flattened and processed by a 3-layers MLP with GELU activations.

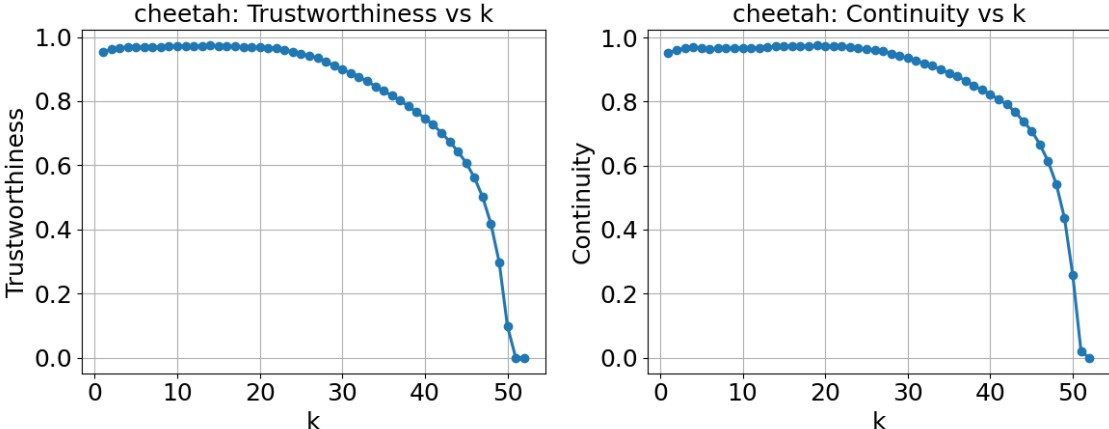

Figure 11: Trustworthiness and continuity as a function of neighborhood size $k$ for MO-HalfCheetah (mean over seeds 0–4).

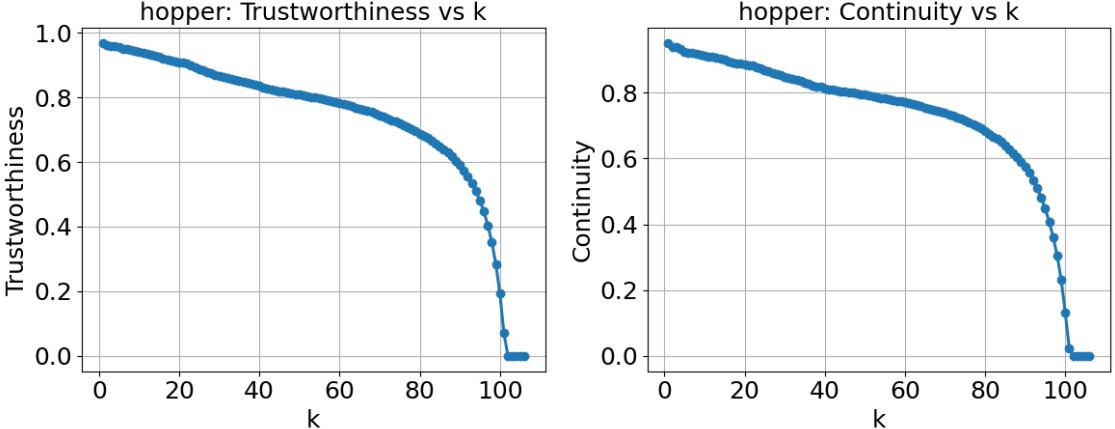

Figure 12: Trustworthiness and continuity as a function of neighborhood size $k$ for MO-Hopper (mean over seeds 0–4).

table 6 reports the trustworthiness and continuity performance on all environments. While under this aspect results are comparable with the encoder used in the main body, the main difference arises when looking at the embedding space and the scatterplots, which show a weaker structure and high instability.

Table 6: Baseline performance measured by trustworthiness and continuity. Higher values indicate better alignment.

| Setting | Comparison | Trustworthiness | Continuity |
|---|---|---|---|
| Left–Right | ME vs. TE (baseline) | $0.93 \pm 0.00$ | $0.93 \pm 0.00$ |
| Smooth | ME vs. TE (baseline) | $0.95 \pm 0.03$ | $0.96 \pm 0.02$ |
| Cheetah | PF vs. Baseline | $0.9825 \pm 0.0021$ | $0.9797 \pm 0.0010$ |
| Hopper (3 obj) | PF vs. Baseline | $0.9681 \pm 0.0023$ | $0.9480 \pm 0.0024$ |

| Dataset | Metric | LSTM |
|---------|--------|------|
| Left-Right | Trustworthiness | $0.400 \pm 0.000$ |
| Left-Right | Continuity | $0.453 \pm 0.016$ |
| Smooth | Trustworthiness | $0.925 \pm 0.023$ |
| Smooth | Continuity | $0.975 \pm 0.008$ |

Table 7: Trustworthiness and Continuity (mean ± std across seeds) for LSTM embeddings.

| Metric | HalfCheetah | Hopper |
|--------|-------------|--------|
| Trustworthiness | $0.976 \pm 0.005$ | $0.967 \pm 0.005$ |
| Continuity | $0.982 \pm 0.003$ | $0.936 \pm 0.003$ |

Table 8: Trustworthiness and Continuity (mean ± std across seeds) for HalfCheetah and Hopper LSTM embeddings.

## C.4 LSTM

This baseline first maps the state and action pair of each timestep $(s_t, a_t)$ through a small MLP, then processes the sequence with a 2-layer LSTM using packed sequences to ignore padding, and finally projects the last hidden state to the low-dimensional CLS embedding. It keeps the output embedding at the same size as the other models, while the recurrent hidden state is the same as for the other models, so the LSTM has enough capacity to serve as a meaningful baseline rather than a tiny bottleneck.

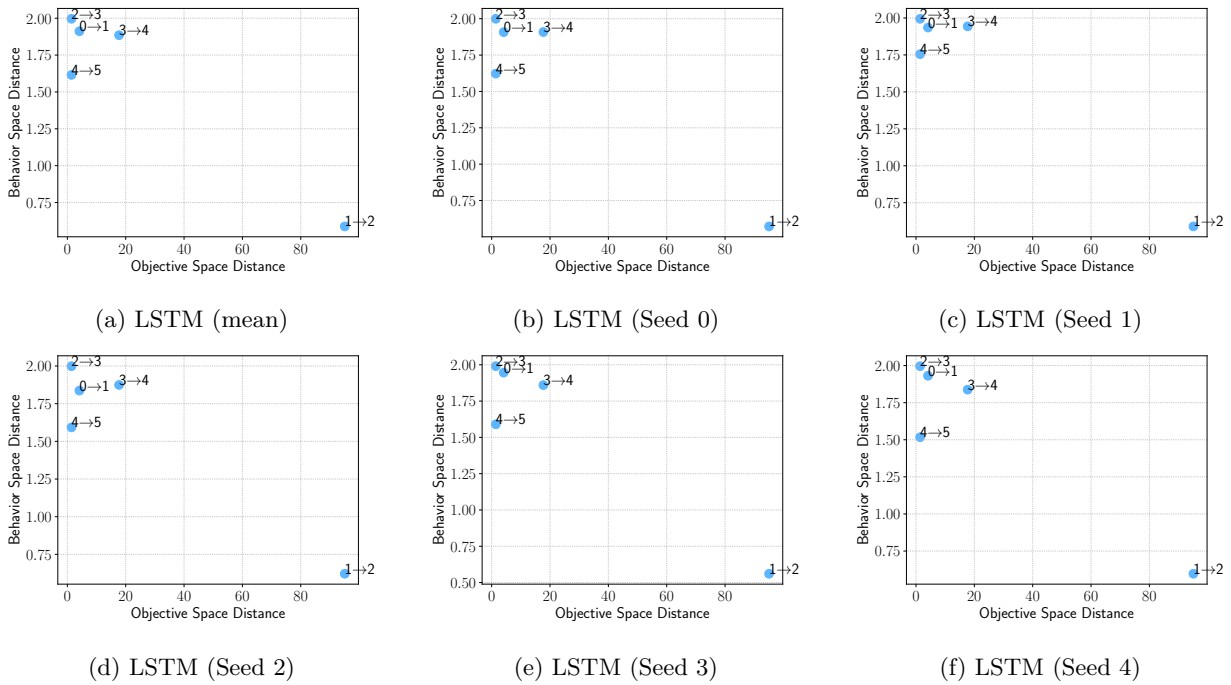

Figure 13: Distances between consecutive policies over the PF in the objective and behavior spaces across different random seeds (0 to 4) for left-right DST, using LSTM embeddings.

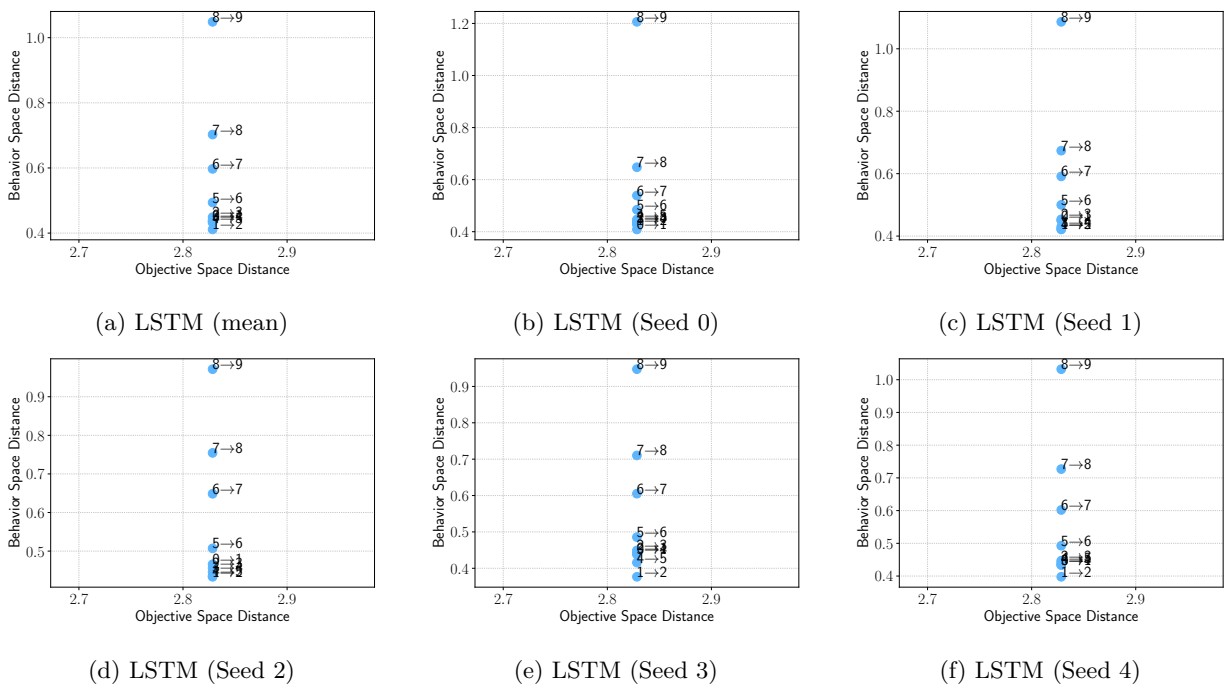

Figure 14: Distances between consecutive policies over the PF in the objective and behavior spaces across different random seeds (0 to 4) for smooth DST, using LSTM embeddings.

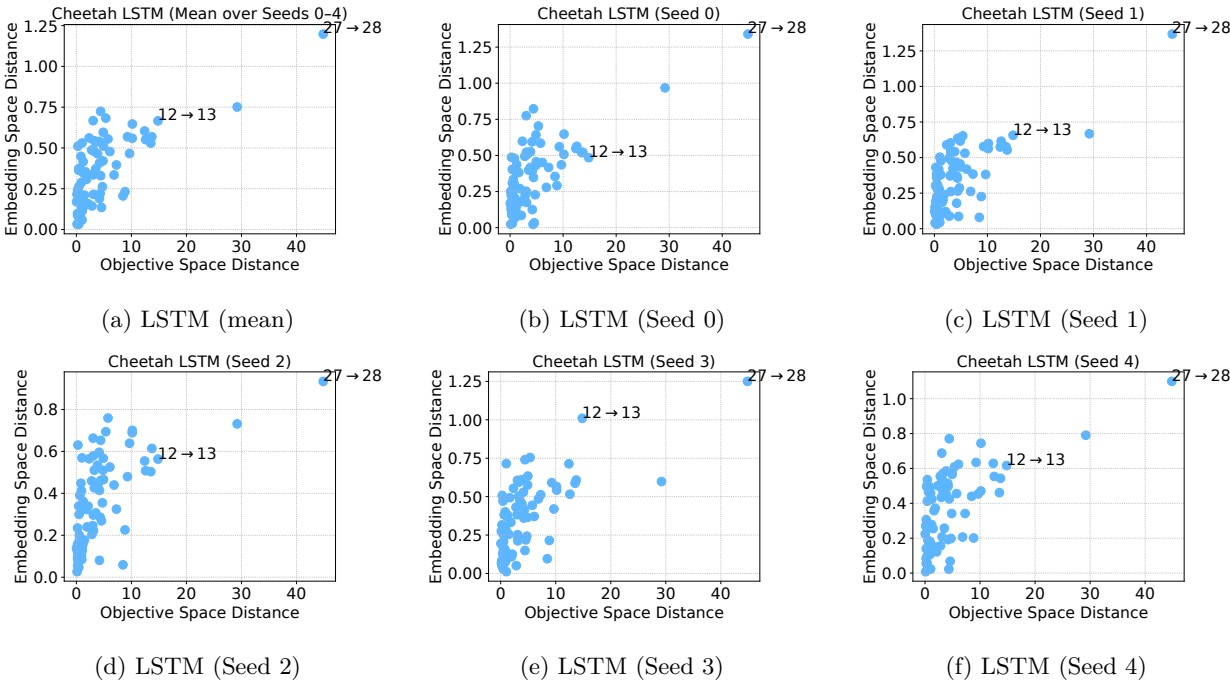

Figure 15: Distances between consecutive policies over the PF in the objective and behavior spaces across different random seeds (0 to 4) for MO-HalfCheetah, using LSTM embeddings.

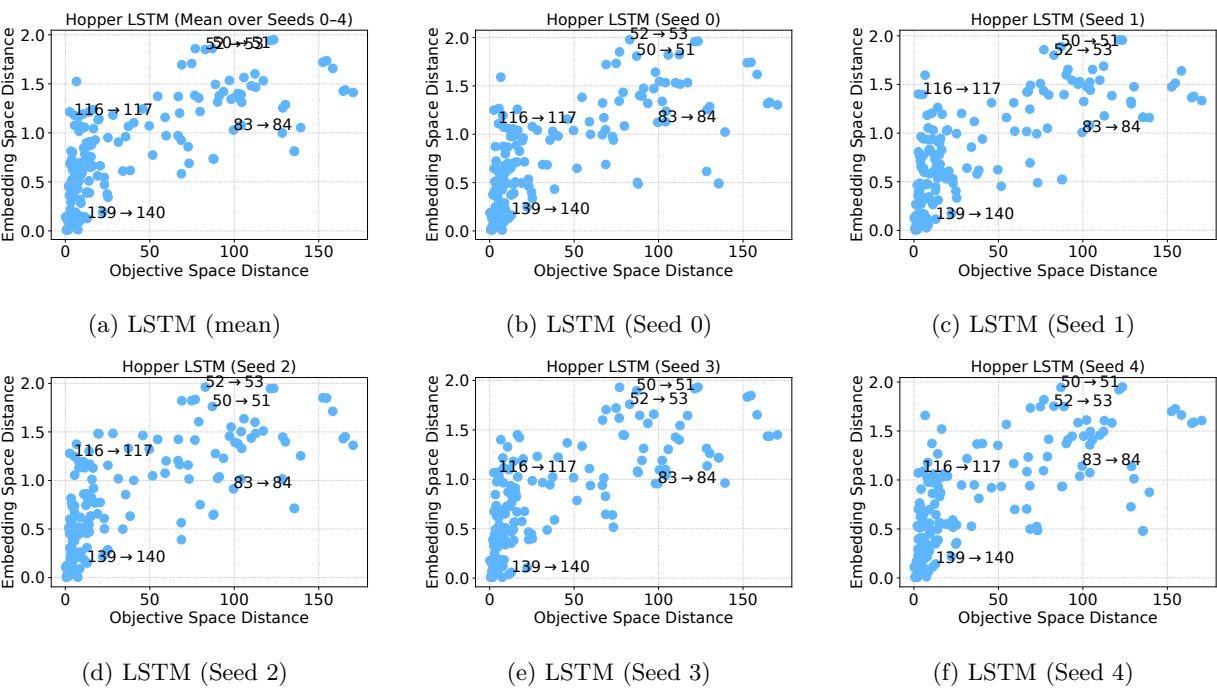

Figure 16: Distances between consecutive policies over the PF in the objective and behavior spaces across different random seeds (0 to 4) for MO-Hopper, using LSTM embeddings.

### C.4.1 Cheetah and Hopper

### C.5 Scatter plots per seeds

## D Policy-on policy video analysis for rendered MuJoCo environments

[2]All videos are available at https://www.youtube.com/@TMLRpaper

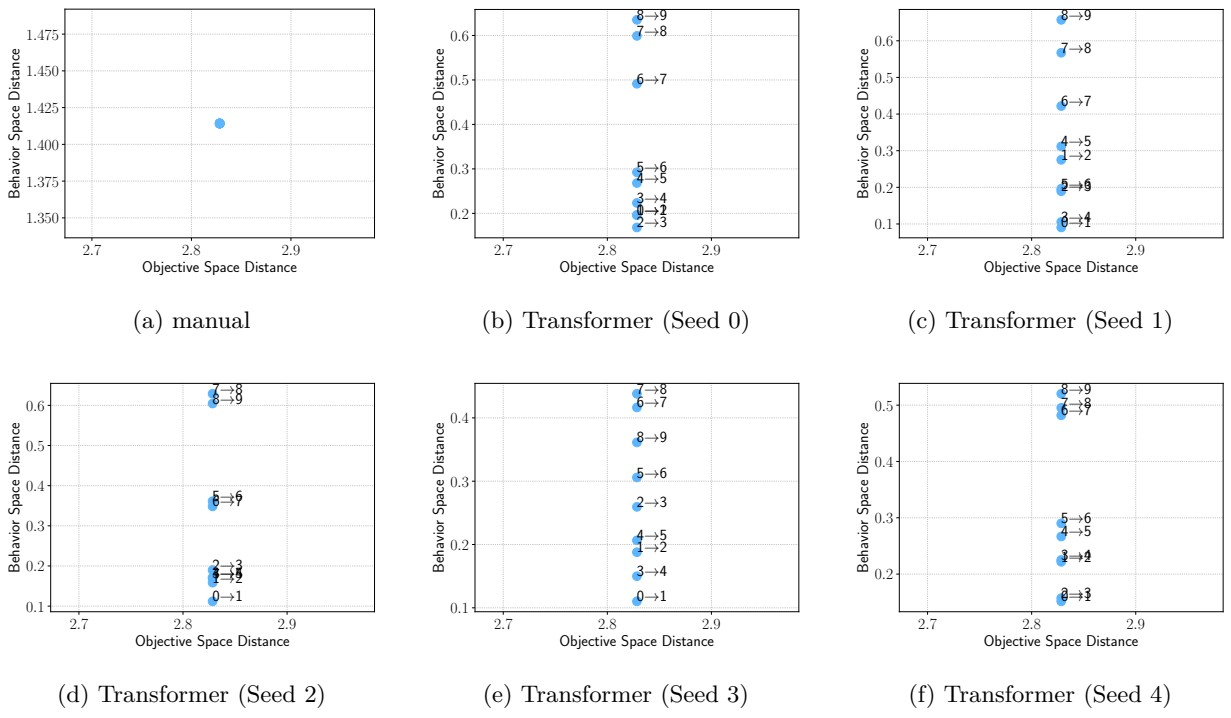

Figure 17: Distances between consecutive policies over the PF in the objective and behavior spaces across different random seeds (0 to 4) for smooth DST.

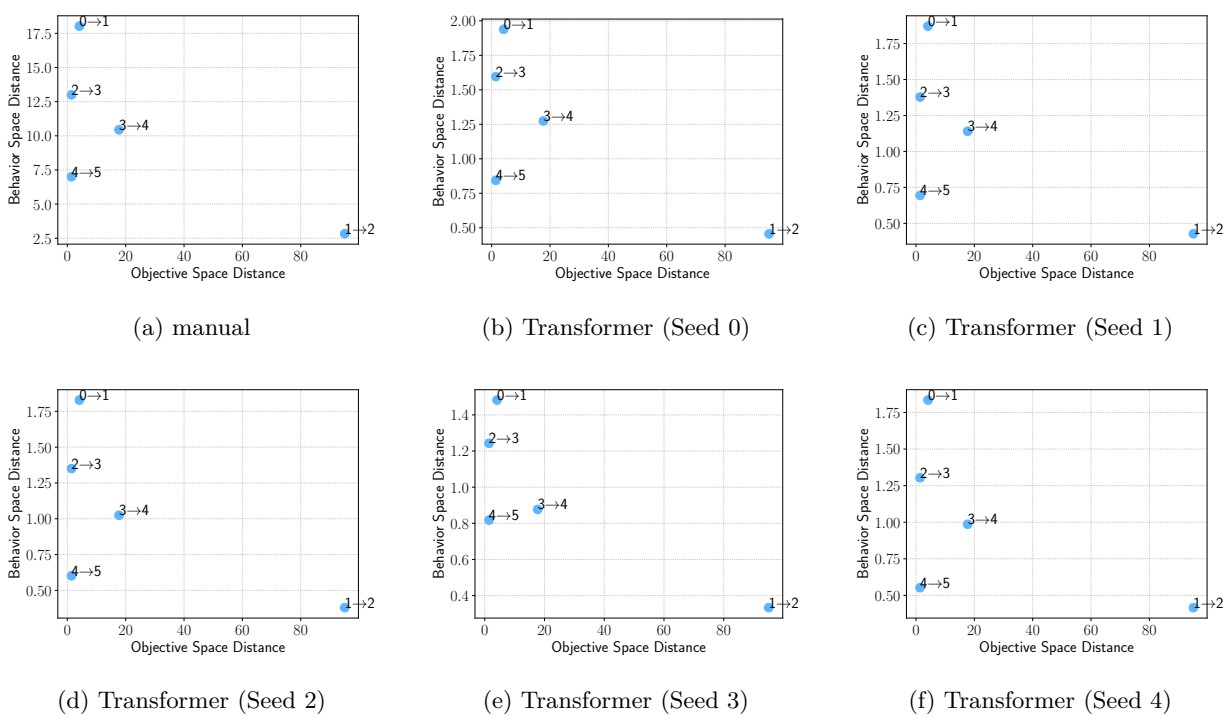

Figure 18: Distances between consecutive policies over the PF in the objective and behavior spaces across different random seeds (0 to 4) for Left–Right DST.

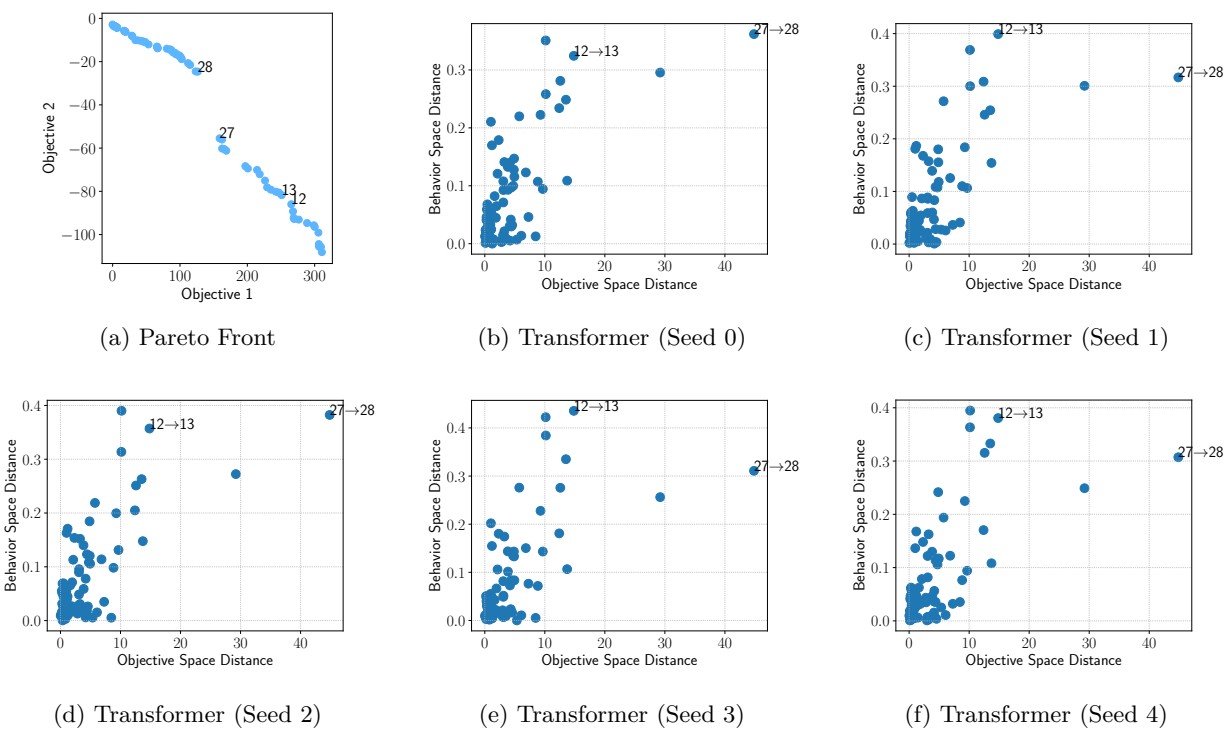

(a) Pareto Front  (b) Transformer (Seed 0)  (c) Transformer (Seed 1)

(d) Transformer (Seed 2)  (e) Transformer (Seed 3)  (f) Transformer (Seed 4)

Figure 19: Distances between consecutive policies over the PF in the objective and behavior spaces across different random seeds (0 to 4).

Table 9: Environment policies grouped into explicit comparison pairs, with behavioral characteristics

| Env. | Policy pair | Policy | Link[2] | Behavioral characteristics |
|------|-------------|--------|---------|---------------------------|
| MO-HalfCheetah | – | policy 12 | Video | – |
| | – | policy 13 | Video | – |
| MO-Hopper | (50,51) | policy 50 | Video | Moves forward with a rigid posture, keeping the body straight and rarely bending the leg joint. |
| | (50,51) | policy 51 | Video | Maintains an upright posture by consistently bending the leg joint during locomotion. |
| MO-Hopper | (52,53) | policy 52 | Video | Raises the front foot and delays forward motion, frequently bending the leg joint before initiating hops. |
| | (52,53) | policy 53 | Video | Plants the front foot forcefully to generate forward push while keeping the leg joints relatively straight. |
| MO-Hopper | (83,84) | policy 83 | Video | Executes aggressive jumps that often lead to loss of balance. |
| | (83,84) | policy 84 | Video | Performs smaller, more conservative movements aimed at preserving balance. |
| MO-Hopper | (116,117) | policy 116 | Video | Advances cautiously through gradual movements due to limited leg joint bending. |
| | (116,117) | policy 117 | Video | Produces high jumps followed by forward falls, characterized by pronounced leg joint bending. |

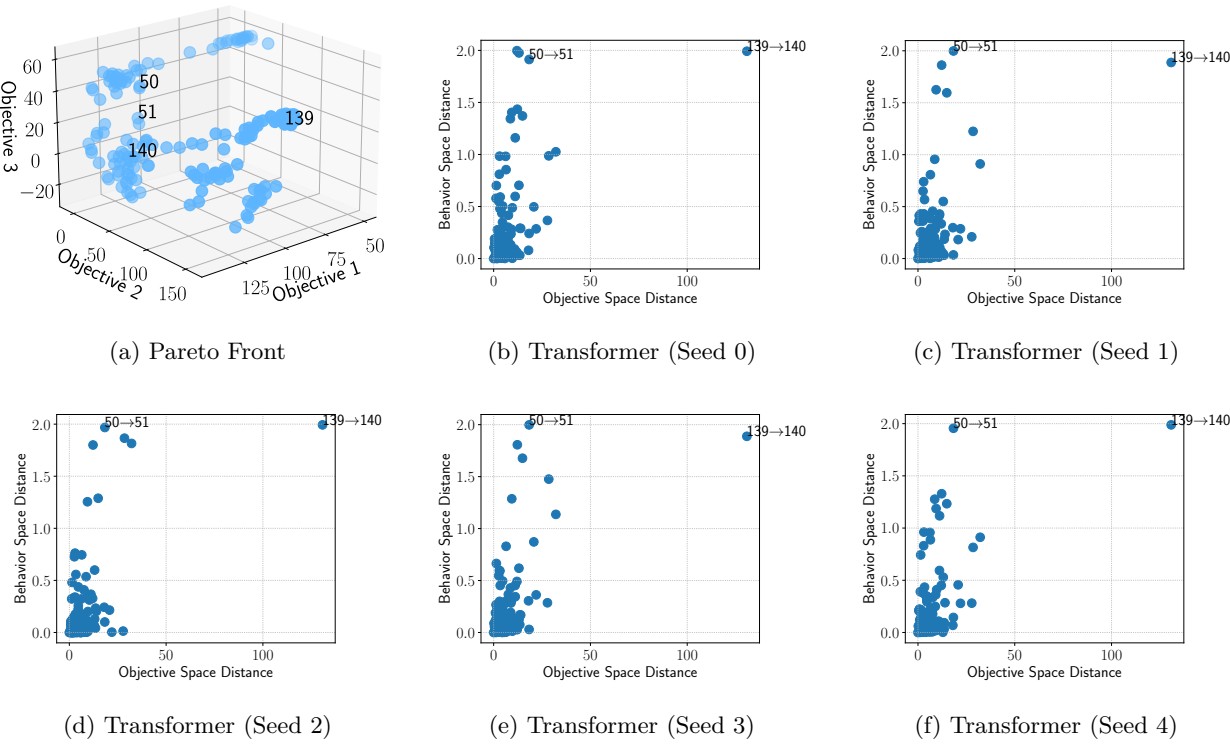

Figure 20: Distances between consecutive policies over the PF in the objective and behavior spaces across different random seeds (0 to 4).

