# OpenReview forum: "Objective-Behavior Alignment: Diagnostics for MORL Policy Selection"
_TMLR — Under review for TMLR_

### Review · Reviewer_RATM · 2026-06-26

**Summary Of Contributions:**

This article focuses on a long overlooked issue in Multi Objective Reinforcement Learning (MORL): Pareto Front only reflects performance trade-offs in the target space and cannot reveal behavioral differences between strategies. The author proposes a model agnostic diagnostic workflow that learns trajectory behavior representation and combines analysis tools such as Trustworthy, Continuity, and Lipschitz inspired scatter plot to analyze the consistency between the target space and behavior space.

**Audience:**

Yes

**Audience Explanation:**

Although this article does not propose a new reinforcement learning algorithm, it focuses on a practical and important issue in MORL - how to analyze and select Pareto policies, rather than just comparing traditional metrics such as Hypervolume.

In recent years, directions such as explainable reinforcement learning, reward design, policy analysis, and human in the loop RL have received increasing attention. Therefore, the diagnostic tool proposed in this paper has certain reference value for researchers engaged in MORL, Decision Support, RL Interpretability, and Reward Engineering. In addition, the paper emphasizes that behavioral space analysis can serve as an important supplement to traditional target space analysis, which has certain novelty and is expected to promote further research on MORL evaluation protocol in the future.

**Broader Impact Concerns:**

This article mainly proposes a diagnostic tool for analyzing MORL strategy behavior, which does not directly change the training process of reinforcement learning strategies, so there is no obvious new ethical risk.
However, as the paper emphasizes the use of behavioral analysis to assist in strategy selection in practical deployment, it is suggested that the author further emphasize that this tool can only detect potential behavioral differences and cannot automatically determine whether these behaviors align with the preferences of real stakeholders, nor can it replace manual decision-making. For safety critical scenarios such as autonomous driving, robot control, etc., the final deployment should still combine domain knowledge and manual review to avoid excessive reliance on diagnostic results.

**Claims And Evidence:**

Yes

**Claims Explanation:**

The core proposition of the paper has received basic experimental support. The author first constructed two scenarios of "consistent behavior" and "inconsistent behavior" using two clearly designed variants of Deep Sea Treasure, verifying that the proposed diagnostic indicators can distinguish between the two; Subsequently, it was extended to the MuJoCo continuous control task, indicating that the method has a certain degree of scalability. In addition, the paper also provides quantitative indicators (Trustworthy, Continuity), local distance analysis (scatter plots), and trajectory visualization, forming a relatively complete experimental demonstration chain.


However, there are still certain limitations to the current evidence.

-Firstly, the paper mainly verifies that diagnostic tools can detect behavioral differences, without further proving that these diagnostic results can truly improve the quality of actual decisions, such as whether they can help users choose strategies that are more in line with their preferences or reduce erroneous decisions.
-Secondly, the experimental scale is still relatively limited, covering only a few MORL benchmarks and mainly relying on a behavior encoder implementation. Due to the high dependence of the entire framework on learned behavior representation, it is currently not sufficient to prove that the framework has stable consistency for different encoders, different MORL algorithms, and more complex environments.

**Requested Changes:**

Critical

1. Strengthen validation of practical usefulness.
    The paper demonstrates that the proposed workflow detects behaviorally different policy pairs, but it remains unclear whether these diagnostics improve downstream policy selection or decision quality. A user study or simulated decision-making experiment would significantly strengthen the practical impact.
2. Evaluate robustness across behavior encoders.
    Since the framework depends heavily on learned behavioral representations, experiments using alternative trajectory encoders or embedding methods would help demonstrate that the conclusions are not specific to a single encoder architecture.
3. Compare against additional behavioral analysis baselines.
    While related work is discussed, experimental comparisons with more existing behavior analysis or policy summarization approaches would better position the proposed workflow relative to prior work.

Non-critical

4. Include sensitivity analysis on key hyperparameters, such as embedding dimension, number of trajectories, neighborhood size k, and aggregation strategy.
5. Provide additional discussion on computational overhead, especially the cost of trajectory encoding for large Pareto sets.
6. Improve the presentation of the scatter plot analysis by providing quantitative thresholds or statistical criteria for identifying “critical regions”, reducing reliance on qualitative interpretation.

---

> ### Author Response · Authors · 2026-07-06
>
> We thank the reviewer for their careful reading of our paper and for the constructive feedback. We are glad the reviewer finds that the work addresses a practical and overlooked issue in MORL. We have uploaded a revised manuscript that addresses the suggested changes. Below, we respond to the limitations and discuss suggested changes raised by the reviewer:
>
> **On strengthening validation of practical usefulness (user study / decision-making experiment).** We agree that demonstrating downstream improvements in decision quality is an important question. However, we believe a full user study is beyond the scope of this work. The gap between objective-space proximity and behavioral similarity is a phenomenon that parts of the MORL community have likely long suspected but that has not previously been surfaced with concrete examples and diagnostic tools. Our primary contribution is to make this issue explicit and actionable. We do this by providing illustrative cases in which the Pareto front is deceptive as a summary of policy similarity, by outlining a workflow for detecting this, and by setting the scene for new research directions. A rigorous user study would require careful experimental design around stakeholder preferences, task framing, and decision protocols; this is a substantial research undertaking that we consider important future work. Measuring decision quality is further complicated by the fact that, as we note in the paper, whether a flagged behavioral difference is deployment-relevant depends on external information about stakeholder preferences that is inherently problem-specific.
> That said, we agree the paper would benefit from a more concrete illustration of how the diagnostics could be used in practice. In the revised manuscript, we added a paragraph to the Conclusion ("An illustrative scenario") describing such a simulated decision-making scenario, which we outline below.
>
> *Revised addition: an illustrative scenario of practical use.* Consider a deployment-oriented reading of the Left–Right DST environment. Suppose the submarine operates in a conflict setting where the map is divided into zones: certain regions (e.g., the left sector) cannot be traversed due to restricted or hostile territory. Ideally, this constraint would be encoded as an additional objective, but in practice the restriction may not have been known at design time or may simply have been overlooked, a failure mode that prior work shows is common even among expert practitioners [1]. In this case, the Pareto front alone gives the decision-maker no indication that some policies traverse the restricted zone while others do not. Our workflow flags exactly these policy pairs: nearly identical in objective space, drastically different in behavior. Upon inspection, the practitioner realizes that a deployment-critical feature was omitted from the problem formulation and can revise the objectives accordingly.
> While in the DST setting this challenge may appear artificial as the environment is a simple synthetic problem designed to make our approach easy to visualize, and the omitted constraint would be trivial to encode once detected, the situation compounds in continuous domains. In MO-Hopper, a practitioner may not know in advance that movement style (e.g., abrupt, large-magnitude actions versus smooth, controlled locomotion) matters for their deployment, and even when aware of it, such behavioral properties are genuinely difficult to formalize as reward terms. Observing behaviorally distinct policies that achieve similar returns helps the practitioner understand the problem itself better: it reveals dimensions of behavioral variation they had not anticipated, and thereby informs how the objectives should be re-defined. This reflects a broader reality of applied (MO)RL: there is exists uncertainty about problem formulation: which objectives matter, and how they should be modeled, is rarely known fully a priori. Our diagnostics support this iterative refinement loop by making visible the behavioral variation that the current formulation does not capture.

---

> ### Author Response · Authors · 2026-07-06
>
> **On experimental scale and encoder dependence**. We thank the reviewer for raising this point and address each aspect in turn.
>
> *Encoder robustness.* We acknowledge that the framework's reliance on learned behavioral representations warrants validation across encoder architectures. The initial manuscript already included an ablation with a simpler MLP baseline encoder (Appendix C.3), showing that our diagnostics remain informative under a different architecture. Following this concern (also raised by reviewer s6t3), we have additionally conducted experiments with an LSTM-based encoder, included in the revised manuscript (Appendix C.4). Across all three architectures (transformer, MLP, LSTM), the diagnostic conclusions are consistent: all encoders yield aligned results under trustworthiness and continuity, supporting the claim that our workflow is not specific to a single encoder implementation. At the same time, the scatter plot analysis shows that on the DST environments, where manual embeddings are available as a reference, the transformer aligns most closely with them, supporting our architectural choice for the main experiments.
>
>
> *Dependence on MORL algorithms*. We would like to clarify that our framework does not depend on the MORL algorithm at all. The workflow operates solely on trajectories and their associated return vectors, regardless of how the policies that produced them were obtained. Policies may come from any MORL algorithm, from analytically derived solutions (as in our DST experiments), or even from non-learning-based methods. We used MORL/D in our experiments simply as one way to generate a Pareto set and the trajectories; the analysis itself is agnostic to it.
>
> *More complex environments.* We agree that broader environment coverage is always desirable. However, our experiments already span a deliberate range of complexity: from tabular grid worlds (DST) to continuous control robotic domains (MO-HalfCheetah, MO-Hopper), scaling in both observation/action dimensionality and in the number of objectives (from two to three). As this is the initial work to investigate objective–behavior misalignment, we prioritized demonstrating the framework on interpretable domains where misalignment can be verified against ground truth, alongside complex domains that establish scalability. We note that visualization and analysis of high-dimensional Pareto fronts (four or more objectives) constitutes a separate research challenge with established solutions [2], independent of our core contribution. We will briefly discuss this in the revised discussion section.
>
> **On comparisons with additional behavioral analysis baselines**. Our contribution is not the embedding technology itself, but the workflow for detecting objective–behavior misalignment in MORL:  a question that, to our knowledge, no prior method directly addresses. This makes direct head-to-head comparisons difficult: existing approaches answer different questions or have incompatible requirements. The closest prior method [3] requires access to Q-values, which excludes the policy sets in our experiments. Other trajectory representation approaches (e.g., TrajCL, Variational Trajectory Embeddings) are embedding methods, not diagnostic workflows; they could serve as alternative encoders within our framework, which is flexible enough to accommodate them. We acknowledge that the initial presentation does not make this positioning sufficiently clear, a point also raised by reviewer 6kZK,  and we revised the Related Work section accordingly.

---

> ### Author Response · Authors · 2026-07-06
>
> On non-critical points:
>
> **On hyperparameter sensitivity analysis.** During development, we examined sensitivity to the listed hyperparameters (embedding dimension, number of trajectories, neighborhood size k, and aggregation strategy) and found that our conclusions were not affected, which is why these analyses were omitted from the submission. Regarding the embedding dimension specifically, we fixed it to 3 across all environments, the only hyperparameter changed from the original encoder work (Mone et al., 2026), so that distances in the behavior space remain meaningful and interpretable, without introducing warps in the space. However, we agree it is a valuable addition and have included this discussion in Appendix A.2. In the revised version, we added a note on the choice of $k$ alongside the definitions of trustworthiness and continuity, and included a sensitivity analysis in Appendix C.2 showing both metrics across the full range of admissible $k$ values, as this parameter most directly influences the reported results. Both metrics are stable across small-to-moderate $k$: for MO-HalfCheetah, values remain above 0.95 up to $k \approx 25$, and for MO-Hopper above 0.8 up to $k \approx 50$; and our reported results ($k=2$) lie well within this stable regime. The decline at large $k$ is expected (larger neighborhoods test increasingly global structure), and the collapse near $k = (2N-1)/3$ is a normalization artifact of the metric definition, which we explain in the appendix.
>
> **On computational overhead.** We added a discussion of computational cost to the revised manuscript (in the Experimental Setup). We note that the workflow scales linearly in the number of policies (each policy contributes a fixed number of rollouts encoded independently), and the encoder itself is lightweight,  in our experiments it was trained on a consumer laptop (MacBook Pro M1).
>
> **On quantitative criteria for critical regions.**  We agree that reducing reliance on qualitative interpretation would strengthen the scatter plot analysis. Deriving principled, generally applicable thresholds is nontrivial, as the relevant distance scales are environment- and encoder-dependent (as our comparison of distance ranges between Cheetah and Hopper illustrates). In our experiments, policy pairs with larger behavior-space distances tended to show visually distinguishable differences in the rendered rollouts, but we refrain from proposing a specific threshold, as more robust criteria are needed.
>
>
> [1] Booth, S., Knox, W. B., Shah, J., Niekum, S., Stone, P., & Allievi, A. (2023). The Perils of Trial-and-Error Reward Design: Misdesign through Overfitting and Invalid Task Specifications. Proceedings of the AAAI Conference on Artificial Intelligence, 37(5), 5920–5929. https://doi.org/10.1609/aaai.v37i5.25733
>
> [2]Osika, Z., Salazar, J. Z., Roijers, D. M., Oliehoek, F. A., & Murukannaiah, P. K. (2023, August). What lies beyond the pareto front? a survey on decision-support methods for multi-objective optimization. In Proceedings of the Thirty-Second International Joint Conference on Artificial Intelligence (pp. 6741-6749).
>
> [3] Osika, Z., Zatarain-Salazar, J., Oliehoek, F. A., & Murukannaiah, P. K. (2024). Navigating Trade-offs: Policy Summarization for Multi-Objective Reinforcement Learning. In ECAI 2024 (pp. 2919-2926). IOS Press.

---

### Review · Reviewer_s6t3 · 2026-06-28

**Summary Of Contributions:**

The authors propose an exploratory diagnostic workflow for Multi-Objective Reinforcement Learning (MORL) to identify "Objective-Behavior misalignment"—a phenomenon where policies that are proximate in the objective space (Pareto front) exhibit significantly divergent behaviors in execution. The workflow utilizes a Transformer-based encoder trained via contrastive learning to compress state-action trajectories into low-dimensional behavioral embeddings. By leveraging neighborhood preservation metrics (Trustworthiness $T(k)$ and Continuity $C(k)$) and Lipschitz-inspired scatter plots, the method provides an automated tool to flag critical policy pairs that require manual inspection prior to real-world deployment.

**Audience:**

Yes

**Audience Explanation:**

The TMLR audience includes a significant subset of researchers focused on reinforcement learning (RL), reward design, and multi-objective optimization. This paper directly addresses a critical and practical issue within these subfields: the risk of scalarized reward functions masking significant, potentially unsafe behavioral variations in trained policies.

**Broader Impact Concerns:**

I do not foresee any negative societal consequences stemming directly from this work. Conversely, by providing a diagnostic tool to expose hidden behavioral variations and potential instabilities in RL policies, this research contributes positively to the safe, transparent, and aligned deployment of autonomous agents in real-world scenarios.

**Claims And Evidence:**

Yes

**Claims Explanation:**

Strengths:

1. High Practical Significance: The paper targets a critical pain point in RL deployment: the fragility of scalarization and the hidden behavioral risks of policies with similar returns. This diagnostic tool enhances policy safety and predictability.
2. Intuitive Motivating Examples: The design of the Left-Right DST versus Smooth DST elegantly isolates and illustrates the core problem of misalignment without unnecessary complexity.

Weaknesses:

1. Lack of Theoretical Guarantees and Bounded Errors: The workflow heavily relies on the empirical approximation of the Transformer encoder. As acknowledged by the authors, even in the perfectly aligned Smooth DST environment, the behavioral space distance fluctuates between $0.2 \sim 0.55$ due to encoder noise. There are no theoretical bounds on this representation error.
2. Limitations of the Ordering Heuristic: The lexicographic ordering heuristic used to sequence policies on the Pareto front for the Lipschitz scatter plots works adequately for 2-objective scenarios but becomes brittle and complex in higher-dimensional spaces (e.g., >2 objectives, as seen in the MO-Hopper environment).

**Requested Changes:**

1. Encoder Robustness and Baseline Expansion: The current ablation study only compares the Transformer against a simple MLP baseline. Please include a comparison with at least one sequence-modeling alternative (e.g., LSTM or VAE-based encoders) to robustly justify the architectural choice of the contrastive Transformer.
2. Hyperparameter Sensitivity Analysis: Provide an analysis of how the core hyperparameters of the behavior encoder (e.g., max_len, hidden dimensions) impact the reliability of the $T(k)$ and $C(k)$ metrics, particularly in long-horizon continuous control tasks.
3. Systematic Threshold for Critical Regions: In the Lipschitz scatter plots, the "critical region" (upper-left) is currently identified via visual inspection. Please add a discussion or heuristic on how practitioners should systematically define the threshold $\rho(i,j)$ to bound false positives in complex, noisy environments.

---

> ### Author Response · Authors · 2026-07-06
>
> We thank the reviewer for their positive assessment and constructive suggestions. We have uploaded a revised version of the paper incorporating the suggested changes, and we address each requested change below.
>
> **1. Encoder robustness (sequence-modeling alternative).** Following the reviewer's suggestion, we have conducted additional experiments with an LSTM-based encoder and include the results in the revised manuscript (in Appendix C.4). The diagnostic conclusions are consistent across all three architectures (transformer, MLP, LSTM), supporting that our workflow does not depend on a specific encoder implementation. At the same time, the scatter plot analysis shows that, on the DST environments where manual embeddings are available as a reference, the transformer is the most robust across seeds and the most closely aligned with them, supporting our architectural choice for the main experiments. Moreover, since encoders are drop-in components in our modular workflow, we expect the quality of the diagnostics to improve as progress is made on trajectory representation learning, with new encoders being directly usable without any change to the rest of the pipeline.
>
> **2. On hyperparameter sensitivity analysis.** During development, we examined sensitivity to the listed hyperparameters (embedding dimension, number of trajectories, neighborhood size k, and aggregation strategy) and found that our conclusions were not affected, which is why these analyses were omitted from the submission. Regarding the embedding dimension specifically, we fixed it to 3 across all environments, the only hyperparameter changed from the original encoder work [1], so that distances in the behavior space remain meaningful and interpretable, without introducing warps in the space. However, we agree it is a valuable addition and have included this discussion in Appendix A.2. In the revised version, we added a note on the choice of $k$ alongside the definitions of trustworthiness and continuity, and included a sensitivity analysis in Appendix C.2 showing both metrics across the full range of admissible $k$ values, as this parameter most directly influences the reported results. Both metrics are stable across small-to-moderate $k$: for MO-HalfCheetah, values remain above 0.95 up to $k \approx 25$, and for MO-Hopper above 0.8 up to $k \approx 50$; and our reported results ($k=2$) lie well within this stable regime. The decline at large $k$ is expected (larger neighborhoods test increasingly global structure), and the collapse near $k = (2N-1)/3$ is a normalization artifact of the metric definition, which we explain in the appendix.
>
> **3. Systematic threshold for critical regions.** We agree that reducing reliance on qualitative interpretation would strengthen the scatter plot analysis. Deriving principled, generally applicable thresholds is nontrivial, as the relevant distance scales are environment- and encoder-dependent (as our comparison of distance ranges between Cheetah and Hopper illustrates). If we had prior assumptions on the application domain (e.g., trading or robotics), additional domain-specific criteria, such as business or safety requirements, could help define meaningful thresholds; however, our workflow is deliberately agnostic to the application. In our experiments, policy pairs with larger behavior-space distances tended to show visually distinguishable differences in the rendered rollouts, but we refrain from proposing a specific threshold, as more robust criteria are needed
>
> Regarding the weaknesses noted: we agree that theoretical bounds on the representation error and a robust policy-ordering method for higher-dimensional objective spaces are important open problems, and both are acknowledged as future work in the paper (Section 5).
>
>
> [1] Antonio Mone, Frans A Oliehoek, and Luciano Cavalcante Siebert. Comi-irl: Contrastive multi-intention inverse reinforcement learning. arXiv preprint arXiv:2602.07496, 2026.

---

### Review · Reviewer_6kZK · 2026-06-30

**Summary Of Contributions:**

This paper aims to provide behavioral insights for multi-objective reinforcement learning (MORL). Specifically, MORL policies with similar value vectors may still exhibit substantial behavioral variation. Such variation has not been sufficiently highlighted in prior work, but it can be critical for decision-makers during policy selection. To address this issue, the paper proposes a diagnostic workflow that combines the behavioral encoder from [1] with several objective-behavior divergence diagnostics, including trustworthiness, continuity, and Lipschitz-style scatter plots. Experimental results on DST variants, namely Left–Right DST and Smooth DST, and MuJoCo environments, namely MO-HalfCheetah and MO-Hopper, show that objective-behavior divergence can exist in certain cases and that the proposed workflow can identify policies with similar value vectors but different behaviors.

[1] Antonio Mone, Frans A Oliehoek, and Luciano Cavalcante Siebert. Comi-irl: Contrastive multi-intention inverse reinforcement learning. arXiv preprint arXiv:2602.07496, 2026.

**Audience:**

Yes

**Audience Explanation:**

Yes. At least some individuals in TMLR’s audience would likely be interested in the findings of this paper, especially researchers working on multi-objective reinforcement learning, reward design, policy selection, and interpretable RL.

The most valuable part of the paper is its emphasis on behavior divergence despite similar objective values. In MORL, policies are typically selected based on their value vectors on the Pareto front. This paper highlights that such value-based summaries can be incomplete: two policies may appear close or nearly equivalent in objective space, while producing substantially different trajectories or action patterns. This is important because a decision-maker may choose a policy believing it represents a small trade-off in values, while in practice it may correspond to a qualitatively different and potentially undesirable behavior.

This finding should be interesting to the TMLR audience because it points to a gap in standard MORL evaluation: Pareto-front quality alone does not necessarily tell us whether policies behave similarly. The proposed diagnostics make this hidden behavioral variation visible and therefore provide useful additional information for policy selection. In my view, this value-behavior divergence is the paper’s strongest and most broadly relevant contribution.

**Broader Impact Concerns:**

I do not see any apparent broader impact concerns. The paper proposes a diagnostic workflow for analyzing behavioral variation in MORL policy selection, and its primary impact is to improve transparency and help decision-makers identify policies whose behaviors may differ despite similar objective values. This could be beneficial for safer and more informed deployment. I do not identify obvious negative societal impacts.

**Claims And Evidence:**

Yes

**Claims Explanation:**

The claims are mostly supported by accurate and clear evidence. The strongest evidence is the controlled DST comparison: Left–Right DST shows low trustworthiness/continuity and clear behavioral divergence, while Smooth DST shows near-perfect alignment. This directly supports the central claim that objective-space proximity may fail to reflect behavioral similarity. The MuJoCo experiments further suggest that the workflow can be applied to continuous-control settings and can flag behaviorally distinct policy pairs, especially in MO-Hopper. However, the evidence is less convincing for broader scalability claims, because the evaluation is limited to a small number of environments, and the interpretability of the learned embedding partially relies on comparison with manually designed embeddings.

**Requested Changes:**

**Main**: The paper should more clearly distinguish its contribution from the prior Behavioral Encoder / CoMI-IRL [1] work. The two papers have some technical overlap in how behavior embeddings are generated, and the current related-work discussion does not sufficiently clarify the difference. The prior work mainly focuses on multi-intention inverse reinforcement learning: it uses contrastive trajectory embeddings to discover behavior modes, cluster demonstrations, and then learn rewards for each cluster. In contrast, this paper focuses on objective-behavior divergence in MORL: it uses behavior embeddings to diagnose whether policies with similar value vectors on the Pareto front may still behave differently. Therefore, the key difference is not primarily the embedding technology itself, but how the technology is used and positioned. Even if the main embedding method is not new relative to the Behavioral Encoder paper, organizing these tools into a workflow for detecting objective-behavior divergence is the main contribution and should be stated more explicitly.

**Minors**:
1. Figure 3 contains too much white space below.
2. Figures 4, 5, 6, and 7 appear somewhat blurry; the authors may consider using vector-format figures, such as PDF, instead of PNG/JPG.
3. The fonts in Figures 4, 5, and 6 are slightly small and could be enlarged for readability.
4.Figure 6(a) is somewhat crowded, and the authors may consider spacing the annotations more clearly or simplifying the labeling.

[1] Antonio Mone, Frans A Oliehoek, and Luciano Cavalcante Siebert. Comi-irl: Contrastive multi-intention inverse reinforcement learning. arXiv preprint arXiv:2602.07496, 2026.

---

> ### Author Response · Authors · 2026-07-06
>
> We thank the reviewer for their careful reading of our paper and for the constructive feedback. We are glad the reviewer finds that the work addresses a practical and overlooked issue in MORL, namely that Pareto-front quality alone does not necessarily indicate whether policies behave similarly, which we make explicit with our proposed diagnostics. We have uploaded a revised manuscript addressing the concerns raised, and we respond to each point below.
>
> Regarding the contribution of our work and its difference from CoMI-IRL, we would like to clarify that our contribution, as the reviewer also noted, is not the embedding technology itself but the proposed workflow for detecting objective–behavior misalignment. We state in the Introduction Section that the contribution is our modular workflow, designed to accommodate for alternative design choices, such as different encoders. Since we agree with the reviewer, we clarified this point further by adding a sentence in the Related Work section of the revised manuscript. Regarding the minor changes to the figures: the white space is caused by the anonymous template but is correct in the non-anonymous version, while the other figures are already in .pdf format.

---

### Author Response · Authors · 2026-07-06

We would like to thank all the reviewers for your feedback and insights. We have tried our best to respond to your questions. We uploaded a revised version of our submission that incorporates your comments.